# World Models in Pieces: Structural Certification for General Agents

Yikai Lu [1]  Yifei Wu [1]  Xinyu Lu [1]  Tongxin Li [1]

## Abstract

In the big-world regime, agents cannot be universally capable and their ability is inevitably specialized across a world model in pieces. Consequently, standard uniform guarantees fail to distinguish between the understanding of critical bottlenecks and irrelevant failures. We first formalize this limitation by proving that *general agents are not universal*, rendering standard worst-case analysis uninformative. To overcome this, we introduce **structural certification**, a transition-local framework that maps bounded goal-conditioned performance to entry-wise guarantees on the agent's internal world model. Our main contribution is constructive. We provide algorithms that filter specific transitions using deep compositional goals and prove that a general agent on these goals has a structural world model with a $\mathcal{O}(1/n) + \mathcal{O}(\delta)$ error bound. Conversely, this bound is tight in the small-$\delta$ regime, whose existence is explicitly guaranteed by our certification. These results enable the certifiable deployment of general agents by localizing the specific transitions where long-horizon planning is reliable.

## 1. Introduction

Reliable long-horizon decision-making requires more than immediate reaction. To plan under a distribution shift and reason counterfactually, an agent must predict how the environment changes under its actions. That is, it needs a causal world model of the dynamics (Sutton, 1990; Richens & Everitt, 2024; Lin et al., 2024; Wang & Huang, 2025). At the same time, general agents are deployed in settings where exhaustive interaction is infeasible and where failures concentrate around a small number of critical bottlenecks, for example, completing realistic web tasks often hinges on a single high-leverage step such as logging in, selecting the

[1]School of Data Science, The Chinese University of Hong Kong, Shenzhen, Shenzhen, China. Correspondence to: Tongxin Li <litongxin@cuhk.edu.cn>.

*Proceedings of the 43rd International Conference on Machine Learning*, Seoul, South Korea. PMLR 306, 2026. Copyright 2026 by the author(s).

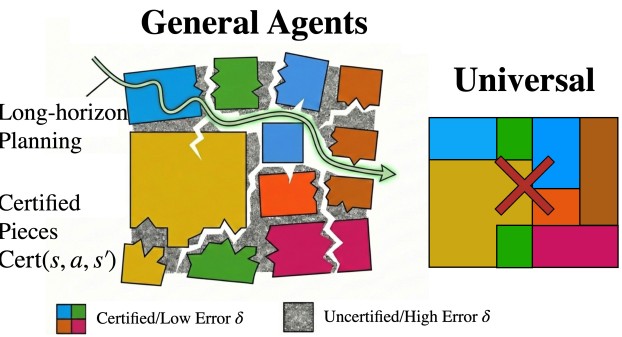

**General Agents**   **Universal**

Long-horizon Planning

Certified Pieces Cert$(s, a, s')$

Certified/Low Error $\delta$   Uncertified/High Error $\delta$

*Figure 1.* **World Models in Pieces.** The capability of a general agent is localized to specific *certified pieces* (colored blocks) with some low error $\delta$ (see Definition 2.3) where its internal model provably aligns with reality. Reliable long-horizon planning (green arrow) succeeds by navigating these certified transitions rather than requiring global accuracy.

correct item variant, or submitting a checkout form (Zhou et al., 2024; Koh et al., 2024). In such settings, overall success is often determined by a small number of decision-critical *bottleneck* transitions, while inaccuracies elsewhere may be irrelevant because they are not relied on along trajectories to success (Frauenknecht et al., 2024; Zhou et al., 2024).

Under the big-world hypothesis, the effective long-horizon goal space is orders of magnitude larger than what any bounded agent can explore with a feasible interaction budget (Javed & Sutton, 2024; Elelimy et al., 2025). Therefore, universal and uniform performance guarantees are not in a reasonable regime, and capability is inevitably uneven across the environment. A general agent may perform well on a set of tasks yet fail on another, and this gap may stay unnoticed until a long-horizon plan relies on those rare or bottleneck transitions (Zhang et al., 2025; Park et al., 2024). It is unrealistic to expect a general agent knows the world model everywhere, i.e., *the general agent cannot be universal*. Therefore, to deploy a general agent on a set of specific goals, the key challenge is to identify which transitions (in other words, what pieces in the world model) the agent needs to know for long-horizon planning, and to certify that it has the required partial knowledge of the world model to achieve the goal.

While a universal assumption allows us to define a single, checkable condition of global success, it fails to capture

the reality of general agents whose capabilities are inherently fragmented across a *world model in pieces*. Recently, (Richens & Everitt, 2024; Richens et al., 2025) show that if the performance of an agent is globally near-optimal over a large task space and a long horizon $n$, then an internal world model must exist and can be recovered from the agent's behavior. While conceptually appealing, certifying general agents requires moving beyond the assumption that the internal model must be globally valid, since a single weak transition can dominate a worst-case guarantee while being irrelevant on most well-performed trajectories (Gadot et al., 2024; Wang et al., 2024c). In a complex environment, an agent interacts with the world in pieces often succeeds despite having a completely wrong model of irrelevant dynamics.

Therefore, in this paper, we consider the following question for general agents:

*Can we structurally certify a general agent for specific tasks when its view of the world is necessarily incomplete?*

## 1.1. Contributions

We affirmatively answer the question above by presenting a transition-local notion of certification that treats an agent's performance as a probe. Instead of assuming that a general agent is near-optimal everywhere, we assume that each general agent is a specialist of handling a subset of goals.

*General agents are not universal.* We first provide an impossibility result (Proposition 3.1), proving that there is no universal agent that maintains a uniform failure rate $\delta$ (see Definition 2.3) over the entire goal space with arbitrarily large goal depth $n$. This formalize the big-world regime considered in this work, necessitating a shift toward structural, transition-focused analysis.

*Structural certification.* Next, we develop a constructive, transition-focused certification process that maps $(\delta, n)$-bounded performance on transition-specific goals to entry-wise alignment of the agent's predictive world model. Our main result (Theorem 3.2) proves that for certified transitions, the agent's behavior uniquely determines a dynamics estimate $\hat{P}_{ss'}(a)$ with an error bound of $\mathcal{O}(1/n) + \mathcal{O}(\delta)$. This provides a strictly tighter guarantee in finite-horizon regimes compared to existing universal analysis.

*Constructive algorithms and fundamental limits.* We provide concrete filtering algorithms (Algorithms 2 and 3) that utilize deep compositional goals to isolate specific transitions and distinguish mastered *pieces* of the world from unreliable heuristics. Conversely, we establish a fundamental limit (Theorem 3.3) showing that outside these certified sets, any construction of a consistent world model from the agents' behaviors must incur a non-trivial mismatch with the true world model dynamics.

These results, visualized in Figure 1, enable the certifiable deployment of general agents by localizing the specific regions where their internal planning is provably reliable. We discuss the related work in Appendix A, and Figure 1 conceptualizes the core idea of this paper.

## 2. Problem Formulation

We use capital letters like $X$ to denote random variables, lower case letters like $x$ to denote a value or state, and bold letters like $\boldsymbol{X}$ for sets throughout this paper.

### 2.1. Environment

We consider a controlled Markov process (cMP) $\mathcal{M} = \{\boldsymbol{S}, \boldsymbol{A}, P_{ss'}(a)\}$ (Sutton & Barto, 2018; Meyn & Tweedie, 2009), which is a Markov decision process with no rewards. An agent takes an action $a \in \boldsymbol{A}$ at state $s \in \boldsymbol{S}$. The consequent transition to the next state $s' \in \boldsymbol{S}$ is governed by the transition probability $P_{ss'}(a)$. We index states and actions by the time step $t$ to denote their position in the sequence. Accordingly, a *trajectory* is defined as a sequence of state-action pairs $\tau = (s_0, a_0, s_1, a_1, \ldots)$ and the *history* is defined as $h_t = (s_0, a_0, \ldots, s_t)$. We use $\pi$ to denote the agent's policy. For simplicity, we make a standard assumption consistent with the literature (Puterman, 2014; Sutton & Barto, 2018).

**Assumption 2.1.** *We assume the cMP environment satisfies: (i) The state space $\boldsymbol{S}$ and the action space $\boldsymbol{A}$ are finite sets, and $|\boldsymbol{A}| \geq 2$. (ii) For all $t \in \mathbb{N}$, $s, s' \in \boldsymbol{S}$ and $a \in \boldsymbol{A}$, the transition probability $P_{ss'}(a) = P(s_{t+1} = s' \mid s_t = s, a_t = a)$. (iii) $\forall s_0, s' \in \boldsymbol{S}$, there always exists a finite sequence of actions to reach $s'$ from $s_0$ with probability $1$.*

With the structural properties of the environment established, we now turn to the specification of complex agent objectives. We use Linear Temporal Logic (LTL) expressions (Pnueli, 1977; Baier & Katoen, 2008) to model the sequential goals we expect an agent to reach. We write $\tau \models \varphi$ to indicate that the trajectory $\tau$ satisfies the LTL formula $\varphi$. Our primitive goal takes the form of $\varphi := \mathcal{O}([\{s, a\} \in \boldsymbol{g}])$, where $\boldsymbol{g} \subseteq \boldsymbol{S} \times \boldsymbol{A}$ and the temporal operator $\mathcal{O}$ is restricted to $\mathcal{O} \in \{\top, \bigcirc, \Diamond\}$, corresponding to *Now*, *Next*, and *Eventually*, where *Now* ($\top$) indicates a goal action must be taken at the current time step $t$; *Next* ($\bigcirc$) expresses a goal state must be reached at the next time step and *Eventually* ($\Diamond$) formulates a goal state must be reached in the future (Pnueli, 1977; Baier & Katoen, 2008). For example, $\varphi = \bigcirc([S = s])$ indicates that the agent must reach state $s$ at the next time step.

To express an ordered sequence of goals, we use $\psi := \langle \varphi_1, \varphi_2, \ldots \rangle$ to indicate the agent must satisfy goal $\varphi_1$ before satisfying $\varphi_2$, and so on (Kress-Gazit et al., 2009). We use $depth(\psi) = n$ to denote the number of ordered goals

in $\psi$ (or temporal horizon (Demri & Schnoebelen, 2002)). Moreover, $\psi = \bigvee_{k=1}^{m} \psi_k$ denotes a composite goal formed as the disjunction of $m$ sequential goals (Vardi & Wolper, 1986). Collecting all possible composite goals into a set $\Psi := \{\psi\}$ as the universal goal set for a given agent and environment, we use $\Psi_n := \{\psi \mid depth(\psi) \leq n\}$ to denote the universal goal set with a maximum depth $n$.

## 2.2. Universal Agents

We are interested in goal-conditioned agents (Liu et al., 2022; Schaul et al., 2015) where policies $\pi$ are determined by histories $h_t$ and goals $\psi$. For simplicity, we here follow the assumption that the environment is fully observed by the agent and the agent follows a deterministic policy (Richens et al., 2025). While $\Psi$ allows arbitrarily complex composite goals, an optimal universal goal-conditioned agent is a policy $\pi(a_t \mid h_t; \psi)$ that maximizes the probability that $\psi$ is reached for all composite goals $\psi$ in the universal goal set $\Psi$. However, it is not realistic to require an agent to be simultaneously universal and optimal (Reichlin et al., 2024). Some previous works relax the performance requirement to satisfy a failure rate $\delta \in [0, 1]$ for a universal goal set $\Psi_n$ with a maximum depth $n$ compared to a universal optimal policy $\pi^*$ (Zhu & Zhang, 2023). Formally, a universal goal-conditioned agent $\pi(a_t \mid h_t; \psi)$ satisfies,

$$P(\tau \models \psi \mid \pi, s_0) \geq (1 - \delta) \max_{\pi} P(\tau \models \psi \mid \pi, s_0), \quad (1)$$

for all $\psi \in \Psi_n$. Here, $\delta$ denotes the uniform failure rate over $\Psi_n$ and $n$ is a maximum goal depth and $s_0$ is an arbitrary initial state.

As shown in (1), the universal bounded goal-conditioned agent assumes that the agent has a uniform failure rate for universal goals. In this scenario, the failure rate can be dominated by a few worst-performing transitions. Empirically, recent studies on LLM-based agents also demonstrate that performance varies significantly across environmental complexities (Li et al., 2025; Xi et al., 2024). Therefore, a universal guarantee is too coarse to define general agents in reality, motivating structural analysis on goal-specific transitions.

**Impossibility of Universal Agents.** We now advance the discussion from practical hurdles to a formal impossibility theorem. Specifically, we prove that a universal agent capable of maintaining a non-trivial performance guarantee uniformly over a universal goal set cannot exist (Proposition 3.1). This finding implies that seeking uniform bounds over universal tasks is futile. We need to shift our perspective from the unattainable universal agents to a more structural paradigm for general agents.

## 2.3. General Agents

General agents cannot handle universal tasks with uniform performance. To tackle this issue, previous approaches rely on temporal-logic specifications to constrain policies for reinforcement learning (Hasanbeig et al., 2019; Voloshin et al., 2022) and employ model checking to analyze learned policies (Gross & Spieker, 2024). However, this does not directly yield which specific transitions the agent models accurately. We therefore focus on constructing goal subsets that isolate a specific transition $(s, a, s')$.

**Definition 2.2** ($(s, a, s')$-specific goal set). *Consider a universal goal set $\Psi_n$ with a maximum depth $n$ and a specific transition $(s, a, s')$ governed by the transition probability $P_{ss'}(a)$. We say a subset $\Psi_n(s, a, s') \subseteq \Psi_n$ is a $(s, a, s')$-specific goal set if and only if for any goal $\psi \in \Psi_n(s, a, s')$, the optimal success probability of $\psi$ is invariant to all transition probabilities except $P_{ss'}(a)$. Formally, $\forall \psi \in \Psi_n(s, a, s')$,*

$$\max_{\pi} P(\tau \models \psi \mid \pi, s_0) = f_{\psi}(P_{ss'}(a)) \quad (2)$$

*holds for some function $f_{\psi}$ that depends exclusively on the transition probability $P_{ss'}(a)$.*

Intuitively, this set of goals can isolate goals for a specific transition $(s, a, s')$, decoupling the optimal success probability from behavior in the rest of the environment, allowing us to shift from universal guarantees (Richens et al., 2025) over $\Psi_n$ to transition-local analysis driven only by $P_{ss'}(a)$. We therefore study performance guarantees on $\Psi_n(s, a, s')$, where the agent can plausibly achieve some better performance (i.e, a smaller failure rate $\delta$), compared with universal analysis (Schaul et al., 2015; Liu et al., 2022).

To formalize such transition-local guarantees, we define the specific transition $(s, a, s')$ in which a general agent reaches a bounded goal-conditioned performance with a fixed failure rate $\delta$ and maximum depth $n$ as below.

**Definition 2.3** ($(\delta, n)$-bounded specific goal set). *Let $(s, a, s')$ be a specific transition and $\Psi_n(s, a, s')$ be its corresponding specific goal set with a maximum depth of $n$. Given an agent $\pi(a_t \mid h_t; \psi)$, we say a set of goals $\Psi_{\delta,n}(s, a, s') \subseteq \Psi_n(s, a, s')$ is a $(\delta, n)$-bounded specific goal set if for all $\psi \in \Psi_{\delta,n}(s, a, s')$, the following inequality holds:*

$$P(\tau \models \psi \mid \pi, s_0) \geq (1 - \delta) \max_{\pi} P(\tau \models \psi \mid \pi, s_0). \quad (3)$$

For a given environment and a given general agent $\pi(a_t \mid h_t; \psi)$, Definition 2.3 defines a class of specific transition goal set for each transition on which a general bounded goal-conditioned agent satisfies a performance guarantee. Based on this, we are interested in whether a general agent's high

performance on a $(\delta, n)$-bounded specific goal set recovers a near-accurate predictive world model.

Hence, rather than relying on a loose $\delta$ over an asymptotic $n$, we investigate whether general agents contain accurate internal representations by demonstrating high capabilities (or small $\delta$) over a small horizon $n$. We provide an explicit construction (see Appendix D) of a non-trivial specific $(\delta, n)$-bounded goal set $\Psi_{\delta,n}(s, a, s')$ with $|\Psi_{\delta,n}(s, a, s')| \geq n$. We will further discuss how to certify agent's performance over this construction of goals and further derive an upper bound for the error of predictive world models $\hat{P}_{ss'}(a)$ implied by agent's behaviors on the constructive goal set with given $\delta$ and $n$ in the following section.

In Table 1 (see Appendix B), we summarize the key differences between our setup and previous works.

## 3. Main Results

Before presenting our main results, we first characterize the impossibility of assuming a universal agent maintains a uniform failure rate $\delta$ over the entire goal space $\Psi_n$ with arbitrarily large goal depth $n$. Our aim is to construct a composite goal $\psi_{\text{fail}}$ where a non-optimal general agent $\pi$ is not bounded goal-conditioned on this goal for any non-trivial uniform failure rate $\delta \in (0, 1)$, which directly demonstrates that pursuing uniform guarantees over universal goals is impossible. We now formalize this construction as follows.

**Proposition 3.1** (General agents are not universal). *Let the environment satisfy Assumption 2.1 and let $\pi$ be any non-optimal deterministic Markovian policy $\pi(a_t \mid h_t; \psi) = \pi(a_t \mid s_t; \psi)$. For any $\delta \in (0, 1)$, there exists a maximum goal depth $N$ and some goal $\psi_{\text{fail}} \in \Psi_N$ such that:*

$$P(\tau \models \psi_{\text{fail}} \mid \pi, s_0) < (1 - \delta) \max_\pi P(\tau \models \psi_{\text{fail}} \mid \pi, s_0).$$

The proof of Proposition 3.1 is given in Appendix C. Proposition 3.1 reveals a fundamental impossibility of seeking universal guarantees in complex environments. However, this negative result does not contradict the existence of general agents, as specialists of a particular set of goals. In other words, a general agent, while not capable of completing all goal-conditioned tasks, may still exhibit optimal or near-optimal behavior on specific problems.

This again necessitates our focus on general agents, but not universal agents. Rather than requiring universal guarantees, we investigate whether a bounded goal-conditioned performance for a subset of goals on a specific transition is sufficient to certify the accuracy of the predictive world model. Specifically, when the agent's performance satisfies a given failure rate $\delta$ on a specific subset of goals $\Psi_{\delta,n}(s, a, s')$ with maximum depth $n$, its predictive world model $\hat{P}_{ss'}(a)$ must necessarily align with reality. The following theorem formalizes this intuition by giving constructive certification

algorithms and a tight alignment bound for the induced transition probability estimate for certified entries.

**Theorem 3.2** (Structural certification). *Let $P_{ss'}(a)$ be the transition probabilities of an environment satisfying Assumption 2.1. Consider an agent with a deterministic policy $\pi$. Fix $\delta \in [0, 0.5)$ and a maximum goal depth $n > 1$. There exist filtering algorithms (for example, Algorithm 1) that filter a specific $(\delta, n)$-bounded goal set with any given $(\delta, n)$.*

*Moreover, for each certified transition $(s, a, s')$, the agent's policy $\pi$ fully determines a transition probability estimate $\hat{P}_{ss'}(a)$ with:*

$$\left| \hat{P}_{ss'}(a) - P_{ss'}(a) \right| \leq \frac{1}{2(n + 1)} \left( \frac{1}{1 - 2\delta} \right)$$
$$+ \frac{\delta}{1 - 2\delta} P_{ss'}(a) \big(1 - P_{ss'}(a)\big).$$

*In particular, when $\delta \ll 1$, the error scales as:*

$$\left| \hat{P}_{ss'}(a) - P_{ss'}(a) \right| = \mathcal{O}\left( \frac{1}{n} \right) + \mathcal{O}(\delta).$$

The proof of Theorem 3.2 is in Appendix D and the structural certification algorithm is shown in Algorithm 1 (see Algorithm 2 and Algorithm 3 in Appendix F for details). Intuitively, the filter constructs a transition-isolating family of probe goals whose pass/fail pattern induces a switching threshold around the true value $P_{ss'}(a)$. By dividing $[0, 1]$ into $n + 1$ levels, the algorithm localizes $P_{ss'}(a)$ between neighboring grid points. This gives an ideal resolution of order $\frac{1}{n+1}$ in the optimal case, and hence an error scale of roughly $\frac{1}{n+1}$. The role of the failure rate $\delta$ is to blur this threshold through bounded sub-optimality, yielding the additional $\mathcal{O}(\delta)$ term in Theorem 3.2.

Theorem 3.2 shows how strong goal-conditioned performance translates into an accuracy guarantee for the agent's predictive world model. It demonstrates that universal performance, especially uniform guarantees on planning over large horizon $n$, are not a necessary condition for determining an accurate predictive world model $\hat{P}_{ss'}(a)$. While planning capabilities are not controllable and often governed by highly non-linear dynamics, the failure rate $\delta$ often collapses through discontinuous leaps rather than predictable evolutions (Janner et al., 2021; Lambert et al., 2020). Therefore, requiring a large horizon $n$ may be a significant challenge. While prior works (Richens et al., 2025) rely on some large horizon $n$ to guarantee the existence of accurate predictive world model since the upper bound of the approximation error scales to $\mathcal{O}(1/\sqrt{n})$, our result shows that an agent could have a precise predictive world model on specific transitions with small failure rate $\delta$ even if horizon $n$ is not that large since our upper bound scales as $\mathcal{O}(1/n)$. This confirms that general agents allow for highly efficient and

**Algorithm 1** Structural Certification for $(s, a, s')$

---

**Require:** Deterministic policy $\pi(\cdot \mid h_t; \psi)$
**Require:** Target specific transition $(s, a, s')$, along with an alternative action $b \neq a$
**Require:** Horizon $n \in \mathbb{N}$, failure rate $\delta \in [0, 0.5)$
**Require:** Anchor transition probability $P_{ss'}(a)$
**Ensure:** Certificate flag $\text{Cert}(s, a, s')$ and estimate $\hat{P}_{ss'}(a)$
 1: Initialize $p_{\min} \leftarrow 0$, $p_{\max} \leftarrow 1$, $r^\star \leftarrow \text{Null}$.
 2: Initialize $\text{Cert}(s, a, s') \leftarrow \text{False}$ and $\hat{P}_{ss'}(a) \leftarrow \text{Null}$.
 3: **Goal construction.** Construct a family of probe goals by repeating the transition pattern $s \xrightarrow{a} s'$ for $n$ trials.
 4: **for** $r = 0$ **to** $n - 1$ **do**
 5:    **Policy query.** Construct two competing probe goals:
 6:    $\psi_a(r, n)$ : exactly $r$ occurrences of $s \xrightarrow{a} s'$.
       $\psi_b(r + 1, n)$ : exactly $r + 1$ occurrences of $s \xrightarrow{a} s'$.
 7:    Query the preferred first action:

$$a_r \leftarrow \arg \max_{x \in \{a, b\}} \pi(x \mid s; \psi_a(r, n) \vee \psi_b(r + 1, n)).$$

 8:    **Certification update.**
 9:    **if** $a_r = a$ **then**
10:       $p_{\max} \leftarrow \min \left\{ p_{\max}, \dfrac{r + 1}{n + 1 - \delta(n - r)} \right\}.$
11:       **if** $r^\star = \text{Null}$ **then**
12:          $r^\star \leftarrow r.$
13:       **end if**
14:    **else**
15:       $p_{\min} \leftarrow \max \left\{ p_{\min}, \dfrac{(r + 1)(1 - \delta)}{n + 1 - (r + 1)\delta} \right\}.$
16:    **end if**
17: **end for**
18: **Certification.** Check whether

$$P_{ss'}(a) \in [p_{\min}, p_{\max}].$$

19: **if** $P_{ss'}(a) \in [p_{\min}, p_{\max}]$ **and** $r^\star \neq \text{Null}$ **then**
20:    $\text{Cert}(s, a, s') \leftarrow \text{True}.$
21:    **Estimate.** Set

$$\hat{P}_{ss'}(a) \leftarrow \frac{r^\star + 0.5}{n + 1}.$$

22: **end if**
23: **return** $\left( \text{Cert}(s, a, s'), \hat{P}_{ss'}(a) \right).$

---

structural identification on some specific transitions with high performance (or small $\delta$).

This result also motivates a constructive approach to performance certification. Accordingly, we consider Algorithm 2 and Algorithm 3 as certification mechanisms that algorithmically certify a specific $(\delta, n)$-bounded goal set for each specific transition. In particular, for each candidate transition $(s, a, s')$ and given parameters $(\delta, n)$, our algorithms

construct a transition-isolating goal set and output a *certificate flag* $\text{Cert}(s, a, s')$ using an anchored transition probability $P_{ss'}(a)$. Conditioning on $\text{Cert}(s, a, s') = \text{True}$, the agent's behavior can uniquely determine an induced estimate (or predictive world model) $\hat{P}_{ss'}(a)$ with the error bound presented in Theorem 3.2.

**Properties of certification algorithms.** The construction of Algorithm 2 and Algorithm 3 involves a filtering step that requires access to the transition probability $P_{ss'}(a)$ or a high-confidence empirical estimate thereof. This requirement is plausible in many settings where a simulator (Ha & Schmidhuber, 2018; Hafner et al., 2020), logged interaction data (Levine et al., 2020), or targeted system identification (Simpkins, 2012) can provide reliable local transition statistics. In order to make the result concise, we here directly use real transition probabilities. Crucially, only filtering requires prior knowledge of $P_{ss'}(a)$, and the subsequent recovery step for determining predictive transition probabilities $\hat{P}_{ss'}(a)$ from the agent's behavior is unsupervised. Hence, it is worth noting that, while Theorem 3.2 yields a tighter upper bound when $\delta$ is small and $n$ is moderate, our goal is not to reconstruct the full environment dynamics. Rather, we aim to characterize the accuracy of the agent's predictive world model on transitions where we can verify $(\delta, n)$-boundedness on the constructed goal sets. Consequently, certified transitions are precisely those that the agent can reliably exploit for compositional planning. In this sense, $P_{ss'}(a)$ is an external reference that anchors certification, rather than an ingredient for model recovery. We will discuss more consequences of Theorem 3.2 in Section 5.

Furthermore, a policy-only approach that infers a dynamics model by simply *rationalizing* the agent's behavior cannot reliably recover the real world model or guarantee alignment with an insufficient goal depth $n$. We define *rationalization* as the construction of a model $\widehat{\mathcal{M}}$ where the policy $\pi$ satisfies the same behavioral specifications on a subset of universal goals as it does in the true environment (e.g., $\pi$ is $(\delta, n)$-bounded). The following theorem formalizes this limitation.

**Theorem 3.3.** *Let* $\hat{\Psi}_n \subseteq \Psi_n$ *be any set of goals with a maximum depth* $n$. *Fix an expected failure rate* $\delta$ *and suppose* $\gamma > \delta$. *Assume* $\pi$ *is not bounded goal-conditioned on* $\hat{\Psi}_n$ *under the true model. Formally, for all* $\psi \in \hat{\Psi}_n$,

$$P(\tau \models \psi \mid \pi, s_0) < (1 - \gamma) \max_\pi P(\tau \models \psi \mid \pi, s_0).$$

*We can construct a world model* $\widehat{\mathcal{M}}$ *that rationalizes the agent's performance on* $\hat{\Psi}_n$. *That is, for all* $\psi \in \hat{\Psi}_n$,

$$P_{\widehat{\mathcal{M}}}(\tau \models \psi \mid \pi, s_0) \geq (1 - \delta) \max_\pi P_{\widehat{\mathcal{M}}}(\tau \models \psi \mid \pi, s_0),$$

*and moreover there exists a transition $(s, a, s')$ such that*

$$\left| P_{\widehat{\mathcal{M}}ss'}(a) - P_{ss'}(a) \right| > \frac{(\gamma - \delta) \max_\pi P(\tau \models \psi \mid \pi, s_0)}{(2 - \delta)n}$$

*where $P_{\widehat{\mathcal{M}}ss'}(a)$ denotes the transition probability of $\widehat{\mathcal{M}}$.*

The proof is provided in Appendix E. Theorem 3.3 shows that when the agent is not bounded goal-conditioned on a reachable goal set $\hat{\Psi}_n$ under the real dynamics, one can construct a recovered model $\widehat{\mathcal{M}}$ that nonetheless renders the same policy bounded performance on $\hat{\Psi}_n$ that differs from the true predictive world model $\mathcal{M}$. Moreover, this difference is not merely an artifact of a particular construction. For *any* model $\widehat{\mathcal{M}}$ that rationalizes $\pi$ on $\hat{\Psi}_n$, the argument in Appendix E implies a necessary non-trivial lower bound (incurring an additional $|\mathbf{S}|$ factor, since the mismatch can be spread across $|\mathbf{S}|$ possible next states). In other words, observing a policy is *not sufficient* to identify the underlying entry-wise world models. To isolate a specific transition probability $P_{ss'}(a)$, we require some external probes as defined in Definition 2.3 and Definition 2.2). Without such external verification, we risk recovering a predictive model that merely rationalizes the observed behavior but fails to match the true dynamics.

**Interpreting upper and lower bounds.** Theorem 3.3 lower bounds a transition-level *dynamics mismatch* between the true model and a rationalized model $\widehat{\mathcal{M}}$. The bound becomes tight when $\delta \ll 1$. This focus on small-$\delta$ is well-justified for general agents, as the primary challenge in model identification often stems from bottleneck states or rare transitions. Meanwhile, the small-$\delta$ condition is guaranteed by our structural certification.

Consider any inference procedure that takes as input only the depth $n$ behavioral specification on $\hat{\Psi}_n$ (e.g., $\pi$ is $(\delta, n)$-bounded on $\hat{\Psi}_n$) and outputs an entry-wise claim for $P_{ss'}(a)$. Any such claim must hold uniformly over all world models compatible with the same input specification. Theorem 3.3 shows that, whenever $\pi$ is not bounded goal-conditioned on $\hat{\Psi}_n$ under the true dynamics, one can construct a model $\widehat{\mathcal{M}}$ that satisfies the same depth-$n$ behavioral constraints on $\hat{\Psi}_n$, yet differs from $\mathcal{M}$ on at least one transition entry by $o(1/n)$.

Hence, no policy-only, behavior-rationalizing recovery algorithm can guarantee in general entry-wise transition accuracy at a uniform rate $o(1/n)$ over behaviorally consistent environments. This is why the leading $\mathcal{O}(1/n)$ term in Theorem 3.2 is tight: it matches the fundamental resolution limit imposed by depth-$n$ behavioral constraints, while the remaining $\mathcal{O}(\delta)$ term quantifies the additional slack permitted by bounded goal-conditioning (vanishing as $\delta \to 0$). Focusing on small $\delta$ on specific transitions for general agents is

therefore a critical verification requirement. Since general agents are typically near-optimal only on localized decision-critical or bottleneck transitions, our *structural certification* (see Theorem 3.2) ensures $\delta$ is small enough for our bounds to provide near-optimal guarantees.

## 4. Experiments

We conduct experiments to verify the effectiveness of our filtering algorithm on identifying transitions where the agent is bounded goal-conditioned on a specific goal set. Specifically, we want to investigate how accuracy of the predictive world model increases as the agent learns to generalize to longer horizon goals on some specific transitions.

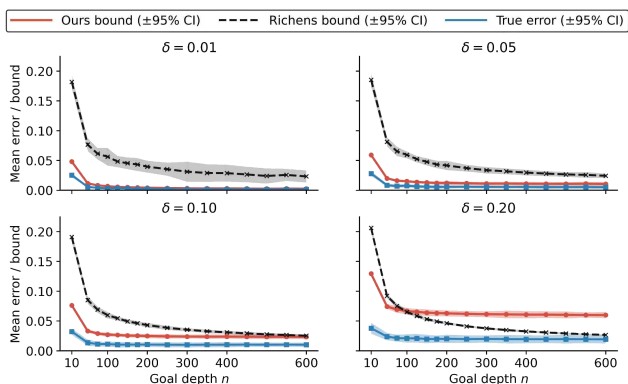

*Figure 2.* For each panel (fixed certification parameter $\delta \in \{0.01, 0.05, 0.10, 0.20\}$), we report the empirical mean recovery error (blue, solid) together with the corresponding certified upper bounds: ours (red, solid) and Richens et al. (2025) (black, dashed), as a function of goal depth $n$. Shaded bands denote $\pm 95\%$ confidence intervals across independently trained agents; for visual clarity, the displayed uncertainty bands (error bars) are magnified by a factor of $3\times$.

**Experimental setup.** We use a stochastic grid world with 20 states and 5 actions ($|\mathbf{S}| = 20, |\mathbf{A}| = 5$). The transition matrix is generated randomly but always follows Assumption 2.1. To provide a concrete analysis, we frame the goal as a key-door navigation task. Specifically, there are some specific transitions corresponding to *picking up a key* and *opening a door* which need precise dynamics knowledge.

To simulate the big world hypothesis, we train 10 general agents independently using random walks with finite samples $N_{\text{samples}}$. We then build an empirical world model for agent using $N_{\text{samples}}$. Crucially, for unvisited transitions $(s, a, s')$, we initialize the empirical world model $\widetilde{P}_{ss'}(a)$ as a uniform distribution. Unless otherwise stated, we will use $N_{\text{samples}} = 4000$ to make sure $\widetilde{P}_{ss'}(a)$ is sparse. The agent's policy $\pi$ is derived by planning an optimal path over its incomplete empirical world model for the LTL goals.

Given parameters $\delta$ and $n$, we use Algorithm 2 and Algorithm 3 to filter the specific $(\delta, n)$-bounded goal set for specific transitions, and approximate the world model to get

$\hat{P}_{ss'}(a)$. We denote empirical mean error as a statistical average of the absolute difference $|\hat{P}_{ss'}(a) - P_{ss'}(a)|$ computed over all certified transition entries collected from 100 randomly selected non-trivial entries $P_{ss'}(a) \in (0.05, 0.95)$. The theoretical bound is a deterministic upper limit derived from Theorem 3.2 and corresponding upper bound from Richens et al. (2025). Importantly, the bound is calculated by directly substituting the fixed failure rate $\delta$ and the specific horizon $n$ into the theoretical formulas, without any averaging. This highlights that our bound is a theoretical guarantee for any transition satisfying the $\delta$ failure constraint, regardless of the specific realization.

**Result analysis.** We first verify the theoretical convergence rate of our error bound. To test the accuracy of approximated world model and tightness of our bound, we here focus on non-trivial transitions entries $P_{ss'}(a)$. Figure 2 reveals the significant advantage of our approach in terms of convergence efficiency. As the goal depth $n$ increases, our certified bound decays linearly, closely tracking the empirical true error. In contrast, the baseline bound remains significantly looser due to its slower $\mathcal{O}(1/\sqrt{n})$ convergence rate, especially when $n$ is small.

To contextualize the practical implication of this numerical gap, consider the key-door scenario in our case study. Successful navigation through bottleneck transitions, for example, unlocking a door with a key requires high-fidelity dynamics knowledge. As shown in Table 2 (see Appendix G), at a planning horizon of $n = 100$, the baseline upper bound overestimates the error to approximately $5.31\%$. In sensitive dynamical systems, this bound-to-error ratio is too coarse to distinguish between a heuristic and genuine physical mastery. In contrast, by tightening the upper bound to a high precision $0.67\%$, our framework certifies that the agent's predictive world model structurally aligns with the environment's true dynamics.

To illustrate the effectiveness of our algorithms, we provide a comprehensive empirical validation for our proposed filtering algorithm Algorithm 2 by comparing the estimation error of certified transitions with the uncertified transitions. Specifically, Figure 3 shows how the Mean Absolute Error $|\hat{P}_{ss'}(a) - P_{ss'}(a)|$ decreases as the maximum goal depth $n$ increases. As expected, the uncertified error (dashed orange line) remains high and constant regardless of $n$. In contrast, the certified error (solid lines) decreases faster as $n$ increases, confirming that our mechanism successfully identifies and makes high-quality estimates. Moreover, tighter confidence failure rates (or smaller $\delta$) result in strictly lower empirical errors. More numerical results are given in Appendix G.

To demonstrate the practical utility of our certification framework, we design two compositional tasks in a larger maze environment $|\boldsymbol{S}| = 625$ to evaluate the reliability of certified transitions. The action set $\boldsymbol{A}$ corresponds to five moves

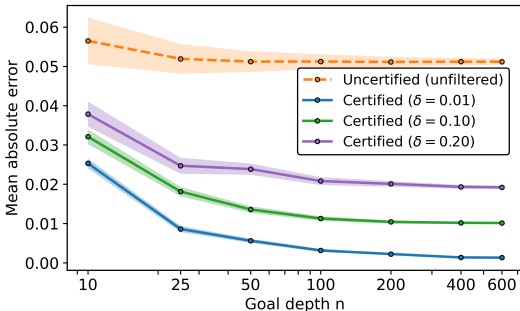

*Figure 3.* **Certified filtering reduces recovery error via goal depth $n$.** We report the mean absolute recovery error $|\hat{P}_{ss'}(a) - P_{ss'}(a)|$ as a function of goal depth $n$, comparing the *uncertified entries* (unfiltered, orange dashed) with *certified entries* obtained under $\delta \in \{0.01, 0.10, 0.20\}$ (solid curves). Markers denote means across independently trained agents and shaded bands indicate $\pm 95\%$ confidence intervals across agents.

the agent can take, which are {UP, DOWN, LEFT, RIGHT, STAY}. Those black cells shown in Figure 4b denote walls, and any action that attempts to cross a wall results in the agent staying in the current cell. We use $\delta = 0.1$ and $n = 100$ to run filtering algorithms and obtain certified transitions. Figure 4a visualizes the filtering results, showing that the procedure identifies transitions on which the predictive model meets a desired accuracy level. In the first task, we position critical bottleneck entities, specifically, a key and a door at transitions successfully certified by our algorithm. In contrast, we placed distinct objects (a pen and paper) at transitions that failed to be certified.

We derive $\pi$ by computing a shortest path on the maze induced by the empirical model $\widetilde{P}_{ss'}(a)$ using $A^*$ algorithm (Hart et al., 1968). The behavioral divergence between these two tasks is striking. As shown in Figure 4b, when guided to unlock the door, the agent could consistently generate near-optimal trajectories, successfully retrieving the key and unlocking the door. However, when guided to write on a paper, the agent exhibits significant erratic behavior and often gets stuck in local loops (see Figure 7 in Appendix G). This validates that our filtering algorithm effectively distinguishes between information mastery and fragile heuristics, thus enabling the safe composition of general agents for long horizon goals. The detailed setup of our case study and more numerical results are provided in Appendix G.

## 5. Conclusion and Discussion

**Impossibility of universal agents.** We establish an obstruction to universal goal-conditioned guarantees. Under Assumption 2.1, Proposition 3.1 shows that any non-optimal agent inevitably violates every non-trivial uniform failure rate once composite goal depth is allowed to be arbitrarily large. This is a fundamentally worst case phenomenon where we define the failure rate $\delta$ via a supremum over

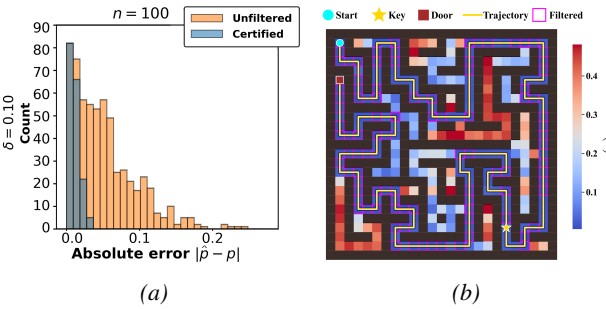

*(a)*           *(b)*

*Figure 4.* **Filtering localizes trustworthy dynamics in a maze.** *(a)* shows a histogram of absolute recovery error $|\hat{P}_{ss'}(a) - P_{ss'}(a)|$ over transition entries at $(\delta, n) = (0.10, 100)$, comparing all recovered entries (unfiltered, orange) with the subset retained by certification (certified, blue). *(b)* visualizes a per-state aggregated error $\epsilon(s)$ (cell color), which sums absolute transition errors over the five actions and local next-state neighborhood (see Appendix G for details). Magenta outlines indicate *certified transitions* and the yellow curve shows *the most frequent trajectory agent generates* from the start (circle) to the key (star) and then to the door (square), illustrating how long-horizon planning routes through certified regions.

the universal goal space $\Psi$ turns a small set of hard goals into a global bottleneck, making uniform guarantees vacuous in large goal space. This mirrors a limitation for uniform performance guarantee across goal classes (Wolpert & Macready, 1997) and the well-known conservatism of mini-max performance guarantees in uncertain Markov Decision Process (MDP) formulations (Iyengar, 2005; Nilim & El Ghaoui, 2005; Xu & Mannor, 2010). Therefore, the universal agent idealization is not merely strong but generically unattainable in the big world hypothesis (Pinon et al., 2025), motivating our shift to general agents whose capability is inherently local and structured (Amortila et al., 2024).

**Structure of internal representations.** World model is crucial for a goal-conditioned agent to exhibit a high performance on long horizon goals, and internal world models are often proposed to explain high planning capabilities for model-free agents (Li et al., 2023; Liu et al., 2025; Saanum et al., 2024). Recently, results of Richens & Everitt (2024); Richens et al. (2025) support the model-based architectures (Hafner et al., 2025; LeCun, 2022) by showing agent policy $\pi$ can determine an approximation of world models with some upper bound. This view is consistent with representation theorems for general agents (Savage, 1972; Halpern & Piermont, 2024), model-based control and learned latent dynamics enable planning and generalization (Ha & Schmidhuber, 2018; Hafner et al., 2025), and analysis of emergent planning mechanisms in systems that are otherwise treated as model-free (Bush et al., 2025; Hao et al., 2023).

Under big world hypothesis, however, such predictive models should not be expected to be uniform across the entire environment (Xu & Mannor, 2010). Instead, the agent is behaviorally constrained to an accurate model only on transitions that it exploits with consistently high capabilities. Beyond these transitions, Theorem 3.3 shows that behavior can provide less constraint (Pinon et al., 2025; Skalse & Abate, 2024) and the implied world model can be recovered inaccurately based on observed behavior without verifications (Bellot et al., 2025; Halpern & Piermont, 2024).

Theorem 3.2 and Algorithm 2 make this intuition operational. When an agent is certified to be $(\delta, n)$-bounded goal-conditioned on a subset of $\Psi_n$, its behavior forces a quantitatively accurate transition estimate on the corresponding certified entries. This yields a sparse set of planning relevant transitions where predictive alignment is provably high-fidelity (Sutton et al., 1999; McGovern & Barto, 2001). We can audit the transitions where a fixed black-box agent has high performance, and therefore, the agent can learn an accurate world model on these transition entries (Bellot et al., 2025; Alshiekh et al., 2018). Intuitively, in a large sparse decision process where only a small number of bottleneck transitions govern long horizon success, its performance typically concentrates on a small set of transitions (McGovern & Barto, 2001; Şimşek & Barto, 2008; Wang et al., 2024a). In our key-door case study, the important transitions are concentrated on those trajectories to find the key and open the door. Hence, the agent is only required to learn world model on those important transitions. Crucially, we do not make any claim on those not certified transitions since we can't guarantee whether the bad performance of observed behavior is caused by planning error or inaccuracy of internal world model. Our certification procedure is designed to identify these transitions and to report where the agent's implied world model is quantitatively accurate.

**Generalization & AI safety.** Our main contribution is a capability boundary characterization for a black-box general agents. Concretely, Theorem 3.2 shows that bounded goal-conditioned performance on a small, verifiable set of LTL goals suffices to certify a subset of transitions. On every certified transition, the agent's behavior determines an implied transition estimate $\hat{P}_{ss'}(a)$ that highly aligns with the true dynamics. In other words, on these specific transitions, high performance enforces predictive alignment.

This enables testing and safe scoping. Instead of relying on universal guarantees or unverifiable claims of uniform performance, our results let us map out a high-performance region, yielding a conservative envelope for deployment in practice. Our certification should be interpreted as a *scoped* audit rather than a black-box guarantee. Accordingly, the resulting safety envelope is conditional: it certifies predictive alignment only on the subset of transitions that is externally verifiable in a trusted testbed (Wu et al., 2024). This is consistent with verification safety pipelines (Banerjee et al., 2024; Miller et al., 2024), which typically require either a known model or conservative bounds on the dynamics

to provide runtime guarantees, and therefore do not apply in fully unknown world deployment without additional assumptions. Moreover, if we believe agent uses its internal representations to plan mentioned above, it can further support safe complex task composition. The transitions certified by Algorithm 2 can serve as reliable sub-goals and we can build more complex compositional goals or constrained objectives on top of them with controllable success probabilities, which is consistent with verification driven Reinforcement Learning and runtime assurance pipelines (Hasanbeig et al., 2019; Voloshin et al., 2022; Shao & Kwiatkowska, 2023; Gross & Spieker, 2024). Consequently, Theorem 3.3 indicates that we can't make any theoretical guarantee for success probabilities outside our certified regions only dependent on agent's behavior (Li et al., 2025). Therefore, applying Theorem 3.2 and Algorithm 2 conservatively determines the subset of transitions on which a general agent is expected to generalize reliably, and thereby enable safe and critical deployment with verifiable boundaries.

**Implications for agent training.** Our results naturally suggest a strategy to train general agents in big worlds. Instead of optimizing for overall performance over universal goals, one can treat the goal as expanding a *certified transition set*: train the agent to achieve $(\delta, n)$-bounded goal-conditioned performance on transition-isolating probe goals (Definitions 2.2 and 2.3), thereby turning demonstrated capabilities into reliable, planning-relevant building sub-goals (Sutton et al., 1999; Machado et al., 2023). In practice, this suggest a training recipe that prioritizes suspected bottleneck transitions (Sunel et al., 2024), where transitions that are likely to control long-horizon success because many trajectories to reach the goal must pass through them. Within those transitions, we have to train the agent more aggressively to fit the predictive world model more accurately.

**Future Directions.** The certification procedure in Algorithm 2 and Algorithm 3 are suited to auditing regimes where one can query the transition probability $P_{ss'}(a)$ for candidate transitions, which enables a supervised filter that identifies a reliable capable domain. Within this domain, Theorem 3.2 shows that demonstrated goal-conditioned performance is sufficient to recover a partial world model with a strictly tighter upper bound than Richens et al. (2025) when failure rate $\delta$ is small and $n$ is moderate. An immediate next step is to generalize the analysis from deterministic goal-conditioned policies to stochastic agents, for example, maximum-entropy and entropy-regularized RL by replacing hard switching logic with likelihood- or occupancy-based tests that certify transitions under randomized action selection, while preserving entry-wise guarantees. In parallel, we plan to relax supervision by developing purely behavioral certification criteria that identify transitions in the agent's capable set without requiring prior access to $P_{ss'}(a)$, and then coupling these criteria with Algorithm 2 and Algorithm 3 to

recover a partial model from logged interaction. Finally, a multi-agent extension is appealing. When different agents possess complementary partial knowledge, one can study principled information exchange protocols , i.e., sharing certified transition information so that agents can expand the set of certified transitions and improve compositional planning by aggregating trustworthy knowledge across individuals.

## Acknowledgment

We sincerely thank all the anonymous area chairs and reviewers for their time and valuable feedback. The first author is grateful to Zihan Huang and Sizhe Li for insightful discussions on the manuscript, and to Yujie Chen for helpful suggestions to improve experiments.

The authors were supported by the Shenzhen Natural Science Foundation in Basic Research Fund under Grant No. J CYJ20250604141200001; the 1+1+1 CUHK–CUHK(SZ)–GDSTC Joint Collaboration Fund No. 2025A0505000047; the Guangdong Basic and Applied Basic Research Foundation No. 2025A1515011311; the National Natural Science Foundation of China (NSFC) No. 72301234; the PengCheng Peacock Scientific Research Fund; the Guangdong Key Lab of Math Foundations for AI No. 2023B1212010001; NSFC No. 62336005; and the Shenzhen Key Lab of Crowd Intelligence Empowered Low-Carbon Energy Network No. ZDS YS202206061006001002.

## Impact Statement

**Ethical considerations.** We believe our framework raises no material ethical concerns.

**Societal implications.** Our transition-local certification framework identifies where a general agent's internal world model is reliably supported by its behavior, with potential implications for AI safety and reliable reinforcement learning.

## A. Related Work

**Predictive world models.** Nowadays, world models are a central component of modern model-based reinforcement learning (Hafner et al., 2025), where agents learn a predictive dynamics model and use it for planning and control (Hansen et al., 2024; Saanum et al., 2024). In complex environments, robust out-of-distribution performance suggests a stricter requirement: the learned agent should capture the causal mechanisms that remain invariant under interventions (Richens & Everitt, 2024). Moreover, Lin et al. (2024) make this view precise by formalizing universal, goal-conditioned robustness. At the same time, many recent works study world model learning as a key driver of memory (Samsami et al., 2024), planning (Hansen et al., 2024), and general control across diverse domains (Hafner et al., 2025), positioning them as the fundamental substrate on which flexible behavior is implemented. Naturally, there is growing interest in whether large language models already function as implicit world models in text, with recent empirical studies supporting the view that sequence models can learn structured latent representations of environments from text (Li et al., 2025).

**General goal-conditioned agents.** Goal-conditioned RL (Schaul et al., 2015) extends the fixed-reward formulation (Sutton & Barto, 2018) by training policies that adapt their behavior based on goals. A major upgrade is to use temporal goals, which rules over a whole horizon rather than a terminal state (Jackermeier & Abate, 2025), for example, tasks like reaching the key then opening the door, often expressed by LTL expressions (Pnueli, 1977). Recent work gives an *optimality-preserving reduction* from LTL goals to limit average rewards via finite-memory reward machines, enabling standard average-reward RL to learn optimal LTL policies, which lets existing RL tools apply more directly. In parallel, a broader perspective views general agents as goal-conditioned systems (Richens et al., 2025) and asks what kind of internal world model is implied by their performance (Richens & Everitt, 2024).

**Bottleneck transitions.** Recent general agents use large language models as controllers (Luo et al., 2025) with tool access (browsers (He et al., 2024), operating systems (Xie et al., 2024), and code execution (Wang et al., 2024b)). Benchmarks have shifted from toy websites to realistic interfaces, including WebArena (Zhou et al., 2024), VisualWebArena (Koh et al., 2024), and OSWorld (Xie et al., 2024). Empirical analyses show that most failures in complex tasks are due to a small number of bottleneck decision-critical steps (Abhyankar et al., 2025), and small errors at those steps can ruin the whole run (Ma et al., 2025). This aligns naturally with temporal goals: many tasks behave like a set of temporal rules, even when the task is specified in natural language (English et al., 2025). It also reframes the world modeling requirement: agents may not require uniformly accurate models, but do require high fidelity on a subset of transitions that determines the task success (Chen et al., 2025).

**Structural certification.** Prior works in RL largely optimize return and report overall performance (or success probabilities) (Sutton & Barto, 2018; Ghasemi et al., 2025), which does not imply identifiable or certified correctness of the underlying dynamics representation (Ng & Russell, 2000; Jin & Syrgkanis, 2024; Zhang & Xie, 2025). Model learning and system identification infer dynamics from data (Yu & Wang, 2024; Zolman et al., 2025; Ding et al., 2025), but they typically aim to fit a global model: they do not explicitly localize which transition entries are actually constrained by the agent's demonstrated behavior, nor which entries are decision-critical for the plans the agent reliably executes (Frauenknecht et al., 2025; Liu & Liu, 2025).

## B. Universal Agents vs. General Agents

*Table 1.* Comparison of Universal vs. General Agents

| Feature | General Agents for Universal Goals (Richens et al., 2025; Richens & Everitt, 2024) | General Agents as Specialists (This Work) |
|---|---|---|
| **Modeling Target** | Worst-case guarantees over universal goal space | **Capability-aware:** Certifies competence on specific transitions |
| **Guarantee Form** | Uniform: $\delta = \sup_{\psi \in \Psi_n} \delta(\psi)$ | **Transition-Specific:** $\delta, n$ defined per certified transition |
| **Model Accuracy** | Dominated by worst-case regions $\Rightarrow$ loose bounds | **Locally Tight:** High fidelity on certified transitions even for finite $n$ |
| **Practicability** | Requires uniform performance guarantees & large horizon $n$ | **Local Sufficiency:** Works with small $\delta$ on subsets of transitions |

We here provide Table 1 to summarize the key differences between our work and existing results.

## C. Proof of Proposition 3.1

In this section, we will prove our warm-up result. For clarity, we will show a proof sketch first.

**Proof Sketch.** Since $\pi$ is non-optimal, there exists a reachable base goal $\psi$ on which $\pi$ has a non-trivial failure rate $\gamma$. When $\delta \leq \gamma$, then $\psi_{fail} := \psi$ naturally gives the desired inequality. Otherwise, we amplify this gap by constructing a depth $N$ sequential goal that requires completing $\psi$ repeatedly, with a return to the same initial state $s_0$ between repetitions. Under a stationary deterministic policy, each repetition contributes a multiplicative failure probability, so the failure rate ratio shrinks to $(1-\gamma)^N$, and we choose sufficiently $N$ large so that $(1-\gamma)^N < 1 - \delta$ which gives the desired inequality.

**Proposition C.1.** *Let the environment satisfy Assumption 2.1 and let $\pi$ be any non-optimal deterministic Markovian policy $\pi(a_t \mid h_t; \psi) = \pi(a_t \mid s_t; \psi)$. For any $\delta \in (0, 1)$, there exists a maximum goal depth $N$ and some goal $\psi_{fail} \in \Psi_N$ such that:*

$$P(\tau \models \psi_{fail} \mid \pi, s_0) < (1 - \delta) \max_{\pi} P(\tau \models \psi_{fail} \mid \pi, s_0).$$

*Proof.* Since $\pi$ is deterministic and non-optimal, there exist some reachable goal $\psi \in \Psi$ and some $\gamma \in (0, 1)$ such that

$$P(\tau \models \psi \mid \pi, s_0) < (1 - \gamma) \max_{\pi} P(\tau \models \psi \mid \pi, s_0). \tag{4}$$

If $\delta \leq \gamma$, pick $\psi = \psi_{\text{fail}}$ and any $N \geq depth(\psi)$ such that $\psi \in \Psi_N$. Then (4) implies

$$P(\tau \models \psi_{\text{fail}} \mid \pi, s_0) < (1 - \gamma) \max_{\pi} P(\tau \models \psi_{\text{fail}} \mid \pi, s_0) \leq (1 - \delta) \max_{\pi} P(\tau \models \psi_{\text{fail}} \mid \pi, s_0),$$

which proves the claim.

Now assume $\delta > \gamma$. Immediately, since $\max_{\pi} P(\tau \models \psi \mid \pi, s_0) > 0$, we have

$$\frac{P(\tau \models \psi \mid \pi, s_0)}{\max_{\pi} P(\tau \models \psi \mid \pi, s_0)} < 1 - \gamma. \tag{5}$$

Fix an integer $N \geq 1$. Under a deterministic policy $\pi$, we define a depth $N$ sequential goal $\psi_{\text{fail}} \in \Psi_N$ as follows: the goal $\psi_{\text{fail}}$ requires completing $\psi$ exactly $N$ times, and between any two consecutive completions it requires returning to $s_0$. The explicit form of this sequential goal $\psi_{\text{fail}}$ is:

$$\psi_{\text{fail}} := \underbrace{\left(\Diamond([S = s_0]) \wedge \psi\right), \left(\Diamond([S = s_0]) \wedge \psi\right), \cdots, \left(\Diamond([S = s_0]) \wedge \psi\right)}_{N \text{ times}}$$

Since this sequential goal needs to be reached by order, by Markovian property of $\pi$,

$$P(\tau \models \psi_{\text{fail}} \mid \pi, s_0) = \left(P(\tau \models \psi \mid \pi, s_0)\right)^N \left(P(\tau \models \Diamond([s = s_0]) \mid \pi, s')\right)^{N-1} \tag{6}$$

since the initial state is $s_0$.

Consider an optimal policy $\pi^\star$ achieving $\max_{\pi} P(\tau \models \psi \mid \pi, s_0)$ for $\psi$ from $s_0$. After each successful completion of $\psi$, an optimal policy $\pi^*$ guarantees the return goal $\Diamond([S = s_0])$ with a success probability no less than any non-optimal policy $\pi$. Therefore,

$$\max_{\pi} P(\tau \models \Diamond([s = s_0]) \mid \pi, s') \geq P(\tau \models \Diamond([s = s_0]) \mid \pi, s'). \tag{7}$$

Combining (6) and (7), we obtain

$$\frac{P(\tau \models \psi_{\text{fail}} \mid \pi, s_0)}{\max_\pi P(\tau \models \psi_{\text{fail}} \mid \pi, s_0)} = \left(\frac{P(\tau \models \psi \mid \pi, s_0)}{\max_\pi P(\tau \models \psi \mid \pi, s_0)}\right)^N \left(\frac{P(\tau \models \Diamond([s = s_0]) \mid \pi, s')}{\max_\pi P(\tau \models \Diamond([s = s_0]) \mid \pi, s')}\right)^{N-1}$$
$$< (1 - \gamma)^N \times 1$$
$$= (1 - \gamma)^N, \tag{8}$$

where the strict inequality uses (5).

Since $\delta > \gamma$, we have $0 < 1 - \delta < 1 - \gamma < 1$, hence $(1 - \gamma)^N \to 0$ as $N \to \infty$. Consequently, there exists an integer $N$ such that

$$(1 - \gamma)^N < 1 - \delta. \tag{9}$$

It suffices to take

$$N \geq \left\lceil \frac{\log(1 - \delta)}{\log(1 - \gamma)} \right\rceil,$$

noting that both logarithms are negative.

With such a choice of $N$, (8) and (9) imply

$$P(\tau \models \psi_{\text{fail}} \mid \pi, s_0) < (1 - \delta) \max_\pi P(\tau \models \psi_{\text{fail}} \mid \pi, s_0),$$

which completes the proof.

$\square$

## D. Proof of Theorem 3.2

We now prove our main theorem Theorem 3.2. We first refer the construction of composite goals provided in Richens et al. (2025).

**Lemma D.1.** *Let $\psi_a(r, n)$ be the composite goal defined as:*

$$\psi_a(r, n) := \langle \varphi_1, \underbrace{\varphi_2, \varphi_3, \varphi_2, \varphi_3, \ldots, \varphi_2, \varphi_3'}_{n \text{ times}} \rangle,$$

*where the agent*

(i) *takes action $A = a$, $\varphi_1 = [A = a]$, and then transitions eventually to $S = s$ and takes action $A = a$, $\varphi_2 = \Diamond([S = s, A = a])$,*

(ii) *transitions next to a goal state which is either $S = s'$, $\varphi_3 = \bigcirc[S = s']$, or $S \neq s'$, $\varphi_3' = \bigcirc[S \neq s']$,*

(iii) *returns eventually to $S = s$ and takes action $A = a$, and repeats the cycle (ii)-(iii) a total of $n$ times, with the transition $\varphi_3 = [S' = s]$ occurring totally $r$ times and the transition $\varphi_3' = [S \neq s']$ occurring totally $n - r$ times.*

*For $s \neq s'$, the optimal policy achieves this goal with probability*

$$\max_\pi P\big(\tau \models \psi_a(r, n) \mid \pi, s_0\big) = \frac{n!}{(n - r)! \, r!} P_{ss'}(a)^r \big(1 - P_{ss'}(a)\big)^{n-r}. \tag{10}$$

Since authors of Richens et al. (2025) have already provided a detailed analysis, we will omit the proof process here.

The above sequential goals $\psi_a(r, n)$ has a goal depth $depth\big(\psi_a(r, n)\big) = 2n + 1$. Consider a sequential goals $\psi_b(r + 1, n)$ identical to $\psi_a(r, n)$ provided in Lemma D.1 except that we require the first sub-goal (i) to take action $A = b$ instead of $A = a$ at time $t = 0$, and in (iii) we require the success times to be $r := r + 1$. We define composite goal $\psi_{a,b}(r, n) := \psi_a(r, n) \vee \psi_b(r + 1, n)$ where $r \in \{0, 1, \ldots, n - 1\}$.

We collect all composite goals $\psi_{a,b}(r, n)$ with a maximum depth $2n + 1$ to a minimal subset $\{\psi_{a,b}(r, n)\}_{r=0}^{n-1} \subseteq \Psi_{2n+1}$. Our aim is to filter the transitions $(s, a, s')$ where the agent could be bounded goal-conditioned on all goals $\psi_{a,b}(r, n) \in \{\psi_{a,b}(r, n)\}_{r=0}^{n-1}$ for a given $n$ and $\delta$.

**Lemma D.2.** *Let the environment satisfy Assumption 2.1 and the agent $\pi$ follows a deterministic policy. Assume transition probability $P_{ss'}(a) \in (0,1)$. If agent is bounded goal-conditioned on any $\psi_{a,b}(r,n) \in \{\psi_{a,b}(r,n)\}_{r=0}^{n-1}$ and we set $r = k$, then the following inequalities hold:*

$$P_{ss'}(a) \begin{cases} \leq \frac{k+1}{n+1-\delta(n-k)}, & \text{if } \pi\big(a_0 \mid s_0; \psi_{a,b}(k,n)\big) = \mathbf{1}([a_0 = a]) \\ \geq \frac{(k+1)(1-\delta)}{n+1-(k+1)\delta}, & \text{if } \pi\big(a_0 \mid s_0; \psi_{a,b}(k,n)\big) = \mathbf{1}\{(a_0 = b)\}. \end{cases} \tag{11}$$

*Proof.* Since $\psi_a(k,n)$ and $\psi_b(k+1,n)$ are mutually exclusive, the optimal success probability for the disjunctive goal $\psi_{a,b}(k,n)$ is:

$$\max_\pi P(\tau \models \psi_{a,b}(k,n) \mid \pi, s_0) = \max\{\max_\pi P\big(\tau \models \psi_a(k,n) \mid \pi, s_0\big), \max_\pi P\big(\tau \models \psi_b(k+1,n) \mid \pi, s_0\big)\}. \tag{12}$$

By definition, when a bounded goal-conditioned agent pursues $\varphi_1 = [A = a]$, or we could say $\psi_a(k,n)$, can achieve success probability $P(\tau \models \psi_a(k,n) \mid \pi, s_0)$. By definition of bounded goal-conditioned agent, applying (1) to (12) and we get:

$$P(\tau \models \psi_a(k,n) \mid \pi, s_0) \geq (1-\delta) \max_\pi P(\tau \models \psi_{a,b}(k,n) \mid \pi, s_0)$$
$$= (1-\delta) \max\{\max_\pi P(\tau \models \psi_a(k,n) \mid \pi, s_0), \max_\pi P(\tau \models \psi_b(k+1,n) \mid \pi, s_0)\}. \tag{13}$$

On the other hand, the agent's success probability for $\psi_a(k,n)$ is upper bounded by the success probability of optimal policy:

$$P(\tau \models \psi_a(k,n) \mid \pi, s_0) \leq \max_\pi P(\tau \models \psi(k,n) \mid \pi, s_0). \tag{14}$$

Combining (13) and (14) yields

$$\max_\pi P(\tau \models \psi(k,n) \mid \pi, s_0) \geq (1-\delta) \max\Big\{ \max_\pi P(\tau \models \psi(k,n) \mid \pi, s_0), \max_\pi P(\tau \models \psi(k+1,n) \mid \pi, s_0)\Big\}$$
$$\geq (1-\delta) \max_\pi P(\tau \models \psi(k+1,n) \mid \pi, s_0). \tag{15}$$

Similarly, if a bounded goal-conditioned agent chooses $\varphi_1 = [A = b]$, then we have

$$\max_\pi P(\tau \models \psi(k+1,n) \mid \pi, s_0) \geq (1-\delta) \max_\pi P(\tau \models \psi(k,n) \mid \pi, s_0). \tag{16}$$

By expanding (15) using Lemma D.1, we get

$$\binom{n}{k} \big(P_{ss'}(a)\big)^k \big(1 - P_{ss'}(a)\big)^{n-k} \geq (1-\delta) \binom{n}{k+1} \big(P_{ss'}(a)\big)^{k+1} \big(1 - P_{ss'}(a)\big)^{n-k-1}. \tag{17}$$

Since $P_{ss'}(a) \in (0,1)$ and $k \in \{0, 1, \ldots, n-1\}$. Here, we can divide (17) by $\big(P_{ss'}(a)\big)^k \big(1 - P_{ss'}(a)\big)^{n-k-1}$,

$$\binom{n}{k} \big(1 - P_{ss'}(a)\big) \geq (1-\delta) \binom{n}{k+1} P_{ss'}(a). \tag{18}$$

Rearranging (18) gives

$$P_{ss'}(a) \leq \frac{k+1}{n+1-\delta(n-k)},$$

which proves the first case of the claim.

When $\varphi_1 = [A = b]$, the agent pursues $\psi_b(k+1, n)$. Similarly,

$$\binom{n}{k+1}\left(P_{ss'}(a)\right)^{k+1}\left(1 - P_{ss'}(a)\right)^{n-k-1} \geq (1-\delta)\binom{n}{k}\left(P_{ss'}(a)\right)^k\left(1 - P_{ss'}(a)\right)^{n-k}. \tag{19}$$

Rearranging (19) gives

$$P_{ss'}(a) \geq \frac{(k+1)(1-\delta)}{n+1-(k+1)\delta},$$

which proves the second case of the claim. Thus, this proves Lemma D.2.

$\square$

*Remark* D.3. Lemma D.2 assumes $P_{ss'}(a) \in (0, 1)$ so that the simplification process is well-defined. To handle the trivial case where $P_{ss'}(a) = \{0, 1\}$, we will use another different goal construction in our algorithms.

Based on Lemma D.2, we first provide an algorithm Algorithm 2 to certify whether an agent is bounded goal-conditioned on $\{\psi_{a,b}(r, n)\}_{r=0}^{n-1}$ when $P_{ss'}(a) \in (0, 1)$. For those transition probabilities $P_{ss'}(a) \in \{0, 1\}$, we construct $\widetilde{\psi}_a(r, n)$ to require $\varphi_2 = \bigcirc[S' = s]$ occurring $k$ times for all $k \leq r$ and $\widetilde{\psi}_b(r, n)$ to require $\varphi_2 = \bigcirc[S' = s]$ occurring $k$ times for all $k > r$. We construct $\widetilde{\psi}_{a,b}(r, n) = \widetilde{\psi}_a(r, n) \vee \widetilde{\psi}_a(r, n)$ (Richens et al., 2025). Applying this set of goals $\{\widetilde{\psi}_{a,b}(r, n)\}_{r=0}^{n}$ to Algorithm 3 gives us a certification for transitions where its corresponding transition probabilities $P_{ss'}(a) \in \{0, 1\}$. Moreover, we can prove that the agent policy $\pi$ fully determines an approximation of world model $\hat{P}_{ss'}(a)$ once the $(\delta, n)$-bounded specific goal set on $(s, a, s')$ is certified. The following theorem details the above statement and provide an upper bound for it.

**Theorem D.4.** *Let $P_{ss'}(a) = P(S_{t+1} = s' \mid A_t = a, S_t = s)$ be the transition probabilities of an environment satisfying Assumption 2.1 and let agent $\pi$ follows a deterministic policy. Fix $\delta \in [0, 0.5)$ and a goal depth $n \in \mathbb{N}$, then there exists an filtering algorithm (see Algorithm 2) that certificates whether a minimal subset $\{\psi_{a,b}(r, n)\}_{r=0}^{n-1} \subseteq \Psi_{2n+1}(s, a, s')$ with a given $(\delta, n)$ is bounded goal-conditioned.*

*Moreover, for each certified transitions $(s, a, s')$, the agent's policy $\pi$ fully determines the a transition probability $\hat{P}_{ss'}(a)$ with:*

$$\left|\hat{P}_{ss'}(a) - P_{ss'}(a)\right| \leq \frac{1}{2(n+1)}\left(\frac{1}{1-2\delta}\right) + \frac{\delta}{1-2\delta}P_{ss'}(a)\left(1 - P_{ss'}(a)\right).$$

*In particular, for highly capable transitions ($\delta \ll 1$), the error of internal representations on the certified transitions scales as:*

$$\left|\hat{P}_{ss'}(a) - P_{ss'}(a)\right| = \mathcal{O}\left(\frac{1}{n}\right) + \mathcal{O}(\delta).$$

*Proof.* When $n$ is sufficiently large in Algorithm 2, the bounded goal-conditioned agent's choices over $\{\psi_{a,b}(r, n)\}_{r=0}^{n-1}$ exhibit a switching point when $P_{ss'}(a) \in (0, 1)$. That is, there exists some $k \in \{1, 2, \ldots, n-1\}$ such that the agent chooses $\varphi_1 = [A = b]$ on $\psi_{a,b}(k-1, n)$ but chooses $\varphi_1 = [A = a]$ on $\psi_{a,b}(k, n)$ for this transition $(s, a, s')$. We fix such switching point an index $k$ when we first observe this switch throughout this proof. For clarity, we denote $p_{min} := \frac{k(1-\delta)}{n+1-k\delta}$ by applying $r = k - 1$ to Lemma D.2 and we denote $p_{max} := \frac{k+1}{n+1-\delta(n-k)}$.

We pick the approximation for the world model as $\hat{P}_{ss'}(a) := \frac{k+0.5}{n+1}$. We can show that $\hat{P}_{ss'}(a) \in [p_{min}, p_{max}]$ since

$$p_{max} - \hat{P}_{ss'}(a) = \frac{k+1}{n+1-\delta(n-k)} - \frac{k+0.5}{n+1} \geq 0,$$

$$\hat{P}_{ss'}(a) - p_{min} = \frac{k+0.5}{n+1} - \frac{(k+1)(1-\delta)}{n+1-(k+1)\delta} \geq 0.$$

By Lemma D.2, $P_{ss'}(a) \in [p_{\min}, p_{\max}]$ when $P_{ss'}(a) \notin \{0, 1\}$. Then, the error of approximation will be bounded by:

$$|\hat{P}_{ss'}(a) - P_{ss'}(a)| \leq \max\{\hat{P}_{ss'}(a) - p_{min}, p_{max} - \hat{P}_{ss'}(a)\} \tag{20}$$

We decompose $\hat{P}_{ss'}(a)$ into $\frac{k+1}{n+1} - \frac{0.5}{n+1}$ to obtain

$$
\begin{aligned}
p_{\max} - \hat{P}_{ss'}(a) &= \frac{k+1}{n+1-\delta(n-k)} - \left( \frac{k+1}{n+1} - \frac{0.5}{n+1} \right) \\
&= \frac{1}{2(n+1)} + \underbrace{\left( \frac{k+1}{n+1-\delta(n-k)} - \frac{k+1}{n+1} \right)}_{\text{Bias}_R}.
\end{aligned}
\tag{21}
$$

Simplifying $\text{Bias}_R$ yields

$$
\begin{aligned}
\text{Bias}_R &= \frac{(k+1)\big[(n+1) - (n+1-\delta(n-k))\big]}{(n+1)(n+1-\delta(n-k))} \\
&= \frac{\delta(k+1)(n-k)}{(n+1)(n+1-\delta(n-k))}.
\end{aligned}
\tag{22}
$$

Since $n - k < n + 1$, the denominator $n + 1 - \delta(n-k) > (n+1)(1-\delta)$. Thus, we can scale (22) to

$$
\begin{aligned}
\text{Bias}_R &\leq \frac{\delta}{1-\delta} \frac{(k+1)(n-k)}{(n+1)^2} \\
&= \frac{\delta}{1-\delta} q_R(1 - q_R), \quad \text{where } q_R = \frac{k+1}{n+1}.
\end{aligned}
\tag{23}
$$

We decompose $\hat{P}_{ss'}(a)$ into $\frac{k}{n+1} + \frac{0.5}{n+1}$ and obtain

$$
\begin{aligned}
\hat{P}_{ss'}(a) - p_{\min} &= \left( \frac{k}{n+1} + \frac{0.5}{n+1} \right) - \frac{k(1-\delta)}{n+1-k\delta} \\
&= \frac{1}{2(n+1)} + \underbrace{\left( \frac{k}{n+1} - \frac{k(1-\delta)}{n+1-k\delta} \right)}_{\text{Bias}_L}.
\end{aligned}
\tag{24}
$$

Simplifying $\text{Bias}_L$,

$$
\begin{aligned}
\text{Bias}_L &= \frac{k\big[(n+1-k\delta) - (n+1)(1-\delta)\big]}{(n+1)(n+1-k\delta)} \\
&= \frac{\delta k(n+1-k)}{(n+1)(n+1-k\delta)}.
\end{aligned}
\tag{25}
$$

Since $k < n + 1$, the denominator $n + 1 - k\delta > (n+1)(1-\delta)$. Thus, we can scale (25) to

$$
\begin{aligned}
\text{Bias}_L &\leq \frac{\delta}{1-\delta} \frac{k(n+1-k)}{(n+1)^2} \\
&= \frac{\delta}{1-\delta} q_L(1 - q_L), \quad \text{where } q_L = \frac{k}{n+1}.
\end{aligned}
\tag{26}
$$

Let $\varepsilon = |\hat{P}_{ss'}(a) - P_{ss'}(a)|$ be the absolute error. Let $f(x) = x(1-x)$ and $C_\delta = \frac{\delta}{1-\delta}$ denote the failure coefficient. We now substitute the discrete variance terms $f(q_R)$ and $f(q_L)$ with the true variance $f\big(P_{ss'}(a)\big)$ the Lipchitz continuity

$$
|f(x) - f(y)| \leq |x - y|.
\tag{27}
$$

Applying (23) to (21), the error is bounded by $\varepsilon \leq \frac{1}{2(n+1)} + C_\delta f(q_R)$, where $q_R = \frac{k+1}{n+1}$. The distance between the discrete point $q_R$ and our estimator $\hat{P}_{ss'}(a)$ is:

$$
|q_R - \hat{P}_{ss'}(a)| = \left| \frac{k+1}{n+1} - \frac{k+0.5}{n+1} \right| = \frac{1}{2(n+1)}.
$$

Using (27) and the triangle inequality,

$$
\begin{aligned}
f(q_R) &\leq f\big(P_{ss'}(a)\big) + |q_R - P_{ss'}(a)| \\
&\leq f\big(P_{ss'}(a)\big) + |q_R - \hat{P}_{ss'}(a)| + |\hat{P}_{ss'}(a) - P_{ss'}(a)| \\
&= f(P_{ss'}(a)) + \frac{1}{2(n+1)} + \varepsilon.
\end{aligned}
\tag{28}
$$

Substituting (28) back into the right-deviation bound implies

$$
\varepsilon \leq \frac{1}{2(n+1)} + C_\delta \left( f\big(P_{ss'}(a)\big) + \frac{1}{2(n+1)} + \varepsilon \right).
\tag{29}
$$

From (26), the error is bounded by $\varepsilon \leq \frac{1}{2(n+1)} + C_\delta f(q_L)$, where $q_L = \frac{k}{n+1}$. The distance between $q_L$ and $\hat{P}$ is symmetric,

$$
|q_L - \hat{P}_{ss'}(a)| = \left| \frac{k}{n+1} - \frac{k+0.5}{n+1} \right| = \frac{1}{2(n+1)}.
$$

Similarly, applying Lipchitz continuity,

$$
\begin{aligned}
f(q_L) &\leq f\big(P_{ss'}(a)\big) + |q_L - P_{ss'}(a)| \\
&\leq f\big(P_{ss'}(a)\big) + |q_L - \hat{P}_{ss'}(a)| + |\hat{P}_{ss'}(a) - P_{ss'}(a)| \\
&= f\big(P_{ss'}(a)\big) + \frac{1}{2(n+1)} + \varepsilon.
\end{aligned}
\tag{30}
$$

Substituting (30) into the left-deviation bound yields an identical inequality to (29) such that

$$
\varepsilon \leq \frac{1}{2(n+1)} + C_\delta \left( f\big(P_{ss'}(a)\big) + \frac{1}{2(n+1)} + \varepsilon \right).
\tag{31}
$$

Since both cases yield the same implicit inequality for $\varepsilon$, we solve for $\varepsilon$ by grouping terms,

$$
\varepsilon(1 - C_\delta) \leq \frac{1}{2(n+1)}(1 + C_\delta) + C_\delta f\big(P_{ss'}(a)\big).
\tag{32}
$$

Substituting $C_\delta = \frac{\delta}{1-\delta}$ back into the coefficient,

$$
1 - C_\delta = 1 - \frac{\delta}{1-\delta} = \frac{1-2\delta}{1-\delta}, \quad 1 + C_\delta = 1 + \frac{\delta}{1-\delta} = \frac{1}{1-\delta}.
$$

Dividing both sides by $(1 - C_\delta)$, if $\delta < 0.5$,

$$
\varepsilon \leq \frac{1}{2(n+1)} \left( \frac{1}{1-\delta} \cdot \frac{1-\delta}{1-2\delta} \right) + \left( \frac{\delta}{1-\delta} \cdot \frac{1-\delta}{1-2\delta} \right) f\big(P_{ss'}(a)\big).
\tag{33}
$$

Simplifying Equation (33) and substituting $f(P_{ss'}(a)) = P_{ss'}(a)\big(1 - P_{ss'}(a)\big)$, we can get the desired result

$$
\varepsilon \leq \frac{1}{2(n+1)} \frac{1}{1-2\delta} + \frac{\delta}{1-2\delta} P_{ss'}(a)\big(1 - P_{ss'}(a)\big).
$$

When $P_{ss'}(a) = \{0, 1\}$, using Algorithm 3 and we can get an unbiased estimation which has been already proved by Richens et al. (2025).

Finally, in the limit of a perfectly rational agent ($\delta \to 0$), the failure rate-induced bias term vanishes, and the recovery error scales linearly with the task horizon, i.e,

$$
|\hat{P}_{ss'}(a) - P_{ss'}(a)| \sim \mathcal{O}(n^{-1}).
$$

This completes the proof. $\qquad\square$

Note that although we used supervised algorithms to filter the high performance transitions, the choice of $k$ and the estimate of predictive world model $\hat{P}_{ss'}(a)$ could be unsupervised once $\{\psi_{a,b}(r,n)\}_{r=0}^{n-1}$ is guaranteed to be $(\delta, n)$-bounded set of goals and $P_{ss'}(a)$ is non-trivial.

# E. Proof of Theorem 3.3

Before we prove Theorem 3.3, we first prove the following auxiliary lemma. Although $\hat{\Psi}_n$ may syntactically allow eventuality operators (e.g., $\Diamond$), such goals can defer satisfaction arbitrarily far into the future. Under our finite interaction perspective, this does not yield additional, entry-wise identifiable constraints on single step transition probabilities beyond what is already witnessed by immediate next-step goals ($\top$ or $\bigcirc$). Therefore, for the purpose of constructing an entry-wise distortion lower bound, it suffices to select goals from the one-step sub-goals $\varphi$ contained in $\Psi_n$. Therefore, it suffices to evaluate trajectory prefixes of length at most $n$.

**Lemma E.1.** *For any goal $\psi \in \Psi_n$ and any deterministic policy $\pi$,*

$$|P(\tau \models \psi \mid \pi, s_0) - P_{\widehat{\mathcal{M}}}(\tau \models \psi \mid \pi, s_0)| \le n\Delta \tag{34}$$

*where $\Delta := \max_{s \in \boldsymbol{S}, a \in \boldsymbol{A}} \mathrm{TV}\big(P(\cdot \mid s, a), P_{\widehat{\mathcal{M}}}(\cdot \mid s, a)\big)$ and $\mathrm{TV}(\mu, \nu) = \frac{1}{2}\|\mu - \nu\|_1$. Here, we use $P(\cdot \mid s, a)$ to denote the distribution over next states.*

*Proof of Lemma E.1.* Fix any $\psi \in \Psi_n$ and policy $\pi$.

By the property of total variation distance, we have

$$\big|P(\tau \models \psi \mid \pi, s_0) - P_{\widehat{\mathcal{M}}}(\tau \models \psi \mid \pi, s_0)\big| \le \mathrm{TV}\Big(P(\tau_{0:n} \in \cdot \mid \pi, s_0), P_{\widehat{\mathcal{M}}}(\tau_{0:n} \in \cdot \mid \pi, s_0)\Big), \tag{35}$$

where we use $P(\tau \in \cdot \mid \pi, s_0)$ to denote the probability measure over trajectory space induced by $\pi$ from $s_0$.

Hence it suffices to show

$$\mathrm{TV}\Big(P(\tau_{0:n} \in \cdot \mid \pi, s_0), P_{\widehat{\mathcal{M}}}(\tau_{0:n} \in \cdot \mid \pi, s_0)\Big) \le n\Delta \tag{36}$$

by (35).

For $t = 0, 1, \ldots, n$, let the history trajectory be $h_t := (s_0, a_0, s_1, a_1, \ldots, s_t)$, and define

$$
\begin{aligned}
d_t &:= \mathrm{TV}\Big(P(\tau_{0:t} \models h_t \in \cdot \mid \pi, s_0), P_{\widehat{\mathcal{M}}}(\tau_{0:t} \models h_t \in \cdot \mid \pi, s_0)\Big) \\
&= \frac{1}{2} \sum_{h_t} \Big|P(\tau_{0:t} \models h_t \mid \pi, s_0) - P_{\widehat{\mathcal{M}}}(\tau_{0:t} \models h_t \mid \pi, s_0)\Big|.
\end{aligned}
\tag{37}
$$

Here, $P(\tau_{0:t} \models h_t \in \cdot \mid \pi, s_0)$ is the probability measure over histories induced by $\pi$.

Since the initial states $s_0$ are the same under both models, $d_0 = 0$.

We claim that for each $t < n$,

$$d_{t+1} \le d_t + \Delta. \tag{38}$$

To prove this, write any $h_{t+1}$ uniquely as $(h_t, a, s')$, where $h_t$ ends at state $s_t$, $a$ is the action at time $t$, and $s'$ is the next state. Under the deterministic oracle, given $h_t$ the action is $a_t = \pi(h_t; \psi)$, so

$$P(\tau_{0:t+1} \models h_{t+1} \mid \pi, s_0) = P(\tau_{0:t} \models h_t \mid \pi, s_0) \sum_{a \in \boldsymbol{A}} \sum_{s' \in \boldsymbol{S}} \mathbf{1}\big\{a = \pi(\psi, h_t)\big\} P_{s_t s'}(a), \tag{39}$$

and similarly, $P_{\widehat{\mathcal{M}}}(\tau_{0:t+1} \models h_{t+1} \mid \pi, s_0) = P_{\widehat{\mathcal{M}}}(\tau_{0:t} \models h_t \mid \pi, s_0) \sum_{a \in \boldsymbol{A}} \sum_{s' \in \boldsymbol{S}} \mathbf{1}\big\{a = \pi(\psi, h_t)\big\} \hat{P}_{s_t s'}(a).$ (40)

where $\mathbf{1}\big\{(a = b)\big\}$ is indicator function taking the value 1 if $a = b$ is true otherwise 0.

Applying (39) and (40) to $d_{t+1}$,

$$
\begin{aligned}
2d_{t+1} &= \sum_{h_{t+1}} \Big|P(\tau_{0:t+1} \models h_{t+1} \mid \pi, s_0) - P_{\widehat{\mathcal{M}}}(\tau_{0:t+1} \models h_{t+1} \mid \pi, s_0)\Big| \\
&= \sum_{h_t} \sum_{a \in \boldsymbol{A}} \sum_{s' \in \boldsymbol{S}} \Big|P(\tau_{0:t} \models h_t \mid \pi, s_0)\mathbf{1}\{a = \pi(\psi, h_t)\}P_{s_t s'}(a) \\
&\qquad\qquad - P_{\widehat{\mathcal{M}}}(\tau_{0:t} \models h_t \mid \pi, s_0)\mathbf{1}\{a = \pi(\psi, h_t)\}\hat{P}_{s_t s'}(a)\Big|.
\end{aligned}
\tag{41}
$$

Insert the intermediate term $P_{\widehat{\mathcal{M}}}(\tau_{0:t} \models h_t \mid \pi, s_0)\mathbf{1}\{a = \pi(\psi, h_t)\}P_{s_t s'}(a)$ and apply the triangle inequality to (41):

$$2d_{t+1} \leq \underbrace{\sum_{h_t} \sum_{a \in \boldsymbol{A}} \sum_{s' \in \boldsymbol{S}} \left| \left(P(\tau_{0:t} \models h_t \mid \pi, s_0) - P_{\widehat{\mathcal{M}}}(\tau_{0:t} \models h_t \mid \pi, s_0)\right)\mathbf{1}\{a = \pi(\psi, h_t)\}P_{s_t s'}(a)\right|}_{(I)}$$

$$+ \underbrace{\sum_{h_t} \sum_{a \in \boldsymbol{A}} \sum_{s' \in \boldsymbol{S}} P_{\widehat{\mathcal{M}}}(\tau_{0:t} \models h_t \mid \pi, s_0)\mathbf{1}\{a = \pi(\psi, h_t)\}\left|P_{s_t s'}(a) - \hat{P}_{s_t s'}(a)\right|}_{(II)}.$$

For (I), using $\sum_{a \in \boldsymbol{A}} \mathbf{1}\{a = \pi(\psi, h_t)\} = 1$ and $\sum_{s' \in \boldsymbol{S}} P_{s_t s'}(a) = 1$,

$$(I) = \sum_{h_t} \left|P(\tau_{0:t} \models h_t \mid \pi, s_0) - P_{\widehat{\mathcal{M}}}(\tau_{0:t} \models h_t \mid \pi, s_0)\right| = 2d_t.$$

For (II), note that for each $(s_t, a)$,

$$\sum_{s' \in \boldsymbol{S}} \left|P_{s_t s'}(a) - \hat{P}_{s_t s'}(a)\right| = 2\,\mathrm{TV}\Big(P(\cdot \mid s_t, a), P_{\widehat{\mathcal{M}}}(\cdot \mid s_t, a)\Big) \leq 2\Delta.$$

Hence

$$(II) \leq \sum_{h_t} P_{\widehat{\mathcal{M}}}(\tau_{0:t} \models h_t \mid \pi, s_0) \sum_{a \in \boldsymbol{A}} \mathbf{1}\{a = \pi(\psi, h_t)\} \cdot 2\Delta = 2\Delta.$$

Combining, we obtain $2d_{t+1} \leq 2d_t + 2\Delta$, i.e. (38).

From $d_0 = 0$ and $d_{t+1} \leq d_t + \Delta$ for $t = 0, \ldots, n-1$, we get by induction $d_n \leq n\Delta$. Since $h_n = (s_0, a_0, \ldots, s_n)$ contains the full length-$n$ trajectory information, we have

$$d_n = \mathrm{TV}\Big(P(\tau_{0:n} \in \cdot \mid \pi, s_0), P_{\widehat{\mathcal{M}}}(\tau_{0:n} \in \cdot \mid \pi, s_0)\Big) \leq n\Delta,$$

which proves (36), thus proves the desired inequality. $\qquad\square$

Then, We present another auxiliary results for the proof,

**Theorem E.2.** *Let $\hat{\Psi}_n \subseteq \Psi_n$ be any set of goals with a maximum depth $n$. Fix an expected failure rate $\delta$ and suppose $\gamma > \delta$. Assume $\pi$ is not bounded goal-conditioned on $\hat{\Psi}_n$ under the true model. Formally, for all $\psi \in \hat{\Psi}_n$,*

$$P(\tau \models \psi \mid \pi, s_0) < (1 - \gamma) \max_\pi P(\tau \models \psi \mid \pi, s_0).$$

*We can construct a world model $\widehat{\mathcal{M}}$ that rationalizes the agent's performance on $\hat{\Psi}_n$. That is, for all $\psi \in \hat{\Psi}_n$,*

$$P_{\widehat{\mathcal{M}}}(\tau \models \psi \mid \pi, s_0) \geq (1 - \delta) \max_\pi P_{\widehat{\mathcal{M}}}(\tau \models \psi \mid \pi, s_0),$$

*and moreover there exists a transition $(s, a, s')$ such that*

$$\left|P_{\widehat{\mathcal{M}} ss'}(a) - P_{ss'}(a)\right| > \frac{(\gamma - \delta) \max_\pi P(\tau \models \psi \mid \pi, s_0)}{(2 - \delta)n}$$

*where $P_{\widehat{\mathcal{M}} ss'}(a)$ denotes the transition probability of $\widehat{\mathcal{M}}$.*

*Proof.* Fix any $\psi \in \hat{\Psi}_n$.

By Lemma E.1, for any policy $\pi$,

$$\begin{aligned} P_{\widehat{\mathcal{M}}}(\tau \models \psi \mid \pi, s_0) &\leq P(\tau \models \psi \mid \pi, s_0) + n\Delta, \\ P_{\widehat{\mathcal{M}}}(\tau \models \psi \mid \pi, s_0) &\geq P(\tau \models \psi \mid \pi, s_0) - n\Delta. \end{aligned}$$

(42)

Taking $\max_\pi$ on both sides gives

$$\max_\pi P_{\widehat{\mathcal{M}}}(\tau \models \psi \mid \pi, s_0) \ \leq \ \max_\pi P(\tau \models \psi \mid \pi, s_0) + n\Delta,$$
$$\max_\pi P_{\widehat{\mathcal{M}}}(\tau \models \psi \mid \pi, s_0) \ \geq \ \max_\pi P(\tau \models \psi \mid \pi, s_0) - n\Delta. \tag{43}$$

Using the assumption of $(1-\delta)$-bounded goal-conditioning of $\pi$ in $\widehat{\mathcal{M}}$, we have

$$P_{\widehat{\mathcal{M}}}(\tau \models \psi \mid \pi, s_0) \ \geq \ (1-\delta)\max_\pi P_{\widehat{\mathcal{M}}}(\tau \models \psi \mid \pi, s_0).$$

Combining and then using (43),

$$\begin{aligned}
P(\tau \models \psi \mid \pi, s_0) &\geq (1-\delta)\max_\pi P_{\widehat{\mathcal{M}}}(\tau \models \psi \mid \pi, s_0) - n\Delta \\
&\geq (1-\delta)\big(\max_\pi P(\tau \models \psi \mid \pi, s_0) - n\Delta\big) - n\Delta \\
&= (1-\delta)\max_\pi P(\tau \models \psi \mid \pi, s_0) - (2-\delta)n\Delta.
\end{aligned}$$

By the assumed non goal-conditioning gap in the true environment,

$$P(\tau \models \psi \mid \pi, s_0) \ < \ (1-\gamma)\max_\pi P(\tau \models \psi \mid \pi, s_0).$$

Therefore,

$$(1-\gamma)\max_\pi P(\tau \models \psi \mid \pi, s_0) \ > \ (1-\delta)\max_\pi P(\tau \models \psi \mid \pi, s_0) - (2-\delta)n\Delta,$$

which rearranges to

$$\Delta \ > \ \frac{(\gamma-\delta)\max_\pi P(\tau \models \psi \mid \pi, s_0)}{(2-\delta)n}. \tag{44}$$

where we recall that $\Delta = \max_{s \in S,\, a \in A} \mathrm{TV}\big(P(\cdot \mid s, a), P_{\widehat{\mathcal{M}}}(\cdot \mid s, a)\big)$. Without additional structural assumptions on how the recovery error is distributed, a worst case can concentrate all discrepancy on a single state-action pair $(s^\dagger, a^\dagger)$. Concretely, we construct

$$P_{\widehat{\mathcal{M}}}(\cdot \mid s, a) = P(\cdot \mid s, a) \quad \forall (s, a) \neq (s^\dagger, a^\dagger), \qquad \mathrm{TV}\big(P(\cdot \mid s^\dagger, a^\dagger), P_{\widehat{\mathcal{M}}}(\cdot \mid s^\dagger, a^\dagger)\big) = \Delta,$$

and the discrepancy inside $(s^\dagger, a^\dagger)$ is a two-point shift. In particular, there exists some $s'^\dagger \in S$ such that

$$\big|\hat{P}_{s^\dagger s'^\dagger}(a^\dagger) - P_{s^\dagger s'^\dagger}(a^\dagger)\big| = \Delta.$$

Thus (44) implies the stated lower bound:

$$\big|\widehat{P}_{ss'}(a) - P_{ss'}(a)\big| \ > \ \frac{(\gamma-\delta)\max_\pi P(\tau \models \psi \mid \pi, s_0)}{(2-\delta)n}.$$

This concludes the proof. $\qquad\square$

## F. Algorithm

In this section, we first present the pseudocode of algorithms that implement the constructive certification in Theorem 3.2. Algorithm 2 handles the non-trivial regime that $P_{ss'}(a) \in (0, 1)$ and Algorithm 3 is introduced to specifically handle trivial cases when $P_{ss'}(a) \in \{0, 1\}$. Both algorithms perform $\mathcal{O}(n)$ iterations with $\mathcal{O}(1)$ work per iteration, so the overall computational complexity is $\mathcal{O}(n)$.

## G. Experiments

Having discussed the detailed setup of our numerical experiments in Section 4, We now introduce the detailed results and our case study of our experiments.

---

**Algorithm 2** Non-trivial Filter and Recover for $(s, a, s')$ if $P_{ss'}(a) \in (0, 1)$

---

**Require:** Deterministic goal-conditioned policy $\pi(a_t \mid h_t; \psi)$
**Require:** Specific transition $(s, a, s')$
**Require:** Fixed horizon $n \in \mathbb{N}$
**Require:** Failure rate $\delta \in [0, 0.5)$.
**Require:** Alternative action $b \neq a$
**Require:** Anchor $P_{ss'}(a) \in (0, 1)$
1: Define certificate flag Cert $\in \{\text{True}, \text{False}\}$
2: Define predictive world model $\hat{P}_{ss'}(a)$
3: Define for $r = \{1, 2, \ldots, n - 1\}$:

$$p_{max}(r) \leftarrow \frac{r + 1}{n + 1 - \delta(n - r)}, \qquad p_{min}(r) \leftarrow \frac{(r + 1)(1 - \delta)}{n + 1 - (r + 1)\delta}.$$

4: Initialize $p_{min} \leftarrow 0$, $p_{max} \leftarrow 1$, $k \leftarrow \text{Null}$, Cert $= \text{False}$, $\hat{P}_{ss'}(a) \leftarrow \text{Null}$
5: **for** $r = 1$ **to** $n - 1$ **do**
6:   Define LTL goals:
     $\varphi_0 \leftarrow [A = a]$; $\varphi_0' \leftarrow [A = b]$
     Require to return to state $s$ and take action $a$
     $\varphi_1 \leftarrow \Diamond[A = a, S = s]$
     Require to return to state $s$ and take action $a$
     $\varphi_2 \leftarrow \bigcirc[S = s']$; $\varphi_2' \leftarrow \bigcirc[S \neq s']$
     Require to reach state $s'$ or not to reach $s'$
     $\psi_1 = \langle \varphi_1, \varphi_2 \rangle$
     Require to return to state $s$, take action $a$ and reach $s'$
7:   $\psi_2 = \langle \varphi_1, \varphi_2' \rangle$
     Require to return to state $s$, take action $a$ and not to reach $s'$
8:   Construct $\psi_a\{r, n\} := \langle \varphi_0, (\psi_1)_{\times r}, (\psi_2)_{\times (n-r)} \rangle$;
     $\psi_b\{r + 1, n\} := \langle \varphi_0, (\psi_1)_{\times (r+1)}, (\psi_2)_{\times (n-r-1)} \rangle$
     Only require $\psi_1$ to occur $r$ or $r + 1$ times and $\psi_2$ to occur $n - r - 1$ times
9:   Construct $\psi_{a,b}(r, n) = \psi_a(r, n) \vee \psi_b(r + 1, n)$.
10:  Query the deterministic initial action:

$$a_r \leftarrow \arg \max_{x \in \{a, b\}} \pi(x \mid s_0; \psi_{a,b}(r, n)).$$

11:  **if** $a_r = a$ **then**
12:    Update upper constraint: $p_{max} \leftarrow \min\{p_{max}, p_{max}(r)\}$.
13:    **if** $k = \text{Null}$ **then**
14:      $k \leftarrow r - 1$
15:    **end if**
16:  **else**
17:    Update lower constraint: $p_{min} \leftarrow \max\{p_{min}, p_{min}(r)\}$.
18:  **end if**
19: **end for**
20: **if** $(p_{min} \leq P_{ss'}(a) \leq p_{max})$ **and** $(k \neq \text{Null})$ **then**
21:  Cert $\leftarrow$ True.
22:  Give the approximation $\hat{P}_{ss'}(a)$ as:

$$\hat{P}_{ss'}(a) \leftarrow \frac{k + 0.5}{n + 1}.$$

23: **end if**
24: **return** $(\text{Cert}, \hat{P}_{ss'}(a))$.

---

---

**Algorithm 3** Trivial Filter and Recover for $(s, a, s')$ when $P_{ss'}(a) \in \{0, 1\}$

---

**Require:** Deterministic goal-conditioned policy $\pi(a_t \mid h_t; \psi)$
**Require:** Specific transition $(s, a, s')$
**Require:** Alternative action $b \neq a$
**Require:** Desired horizon $n \in \mathbb{N}$
**Require:** Anchor $P_{ss'}(a) \in \{0, 1\}$
1: Define certificate flag Cert $\leftarrow \{True, False\}$
2: Define predictive world model $\hat{P}_{ss'}(a) \in \{0, 1\}$
3: Initialize Cert $=$ True and $\hat{P}_{ss'}(a) =$ Null
4: **for** $r = 0$ **to** $n - 1$ **do**
5:     Define $k := r$
6:     Define LTL goals:
        $\varphi_0 \leftarrow [A = a]; \varphi_0' \leftarrow [A = b]$
        Require to take action $a$ or action $b$ at initial time step
        $\varphi_1 \leftarrow \Diamond[A = a, S = s]$
        Require to return to state $s$ and take action $a$
        $\varphi_2 \leftarrow \bigcirc[S = s']; \varphi_2' \leftarrow \bigcirc[S \neq s']$
        Require to reach state $s'$ or not to reach $s'$
        $\psi_1 = \langle \varphi_1, \varphi_2 \rangle$
        Require to return to state $s$, take action $a$ and reach $s'$
7:     $\psi_2 = \langle \varphi_1, \varphi_2' \rangle$
        Require to return to state $s$, take action $a$ and not to reach $s'$
8:     Construct $\widetilde{\psi}_a\{k, n\} := \bigvee_{\text{sequences with } r \leq k \text{ success}} \langle \varphi_0, (\psi_1)_{\times r} (\psi_2)_{\times (n-r)} \rangle$;
        $\widetilde{\psi}_b\{r, n\} := \bigvee_{\text{sequences with } r > k \text{ success}} \langle \varphi_0, (\psi_1)_{\times (r+1)}, (\psi_2)_{\times (n-r+1)} \rangle$
        Only require $\psi_1$ to occur $r$ or $r + 1$ times and $\psi_2$ to occur $n - r - 1$ times
9:     Construct $\widetilde{\psi}_{a,b}(r, n) = \widetilde{\psi}_a(r, n) \vee \widetilde{\psi}_b(r + 1, n)$.
10:    Query the deterministic initial action:

$$a_r \leftarrow \arg\max_{x \in \{a, b\}} \pi(x \mid s_0; \widetilde{\psi}_{a,b}(r, n)).$$

11:    **if** $P_{ss'}(a) = 0$ **and** $a_r = b$ **then**
12:        Cert $\leftarrow$ False.
13:        **break**
14:    **end if**
15:    **if** $P_{ss'}(a) = 1$ **and** $a_r = a$ **then**
16:        Cert $\leftarrow$ False.
17:        **break**
18:    **end if**
19: **end for**
20: **if** Cert $=$ True **then**
21:     $\hat{P}_{ss'}(a) \leftarrow P_{ss'}(a)$
        Output exactly 0 or 1
22: **end if**
23: **return** $(\text{Cert}, \hat{P}_{ss'}(a))$.

---

**Effectiveness of filtering algorithms.** We first provide Table 2 to present a quantitative comparison of the bound tightness. We define tightness gap as the difference between the calculated theoretical upper bound and the actual empirical estimation error (i.e., Gap $=$ Bound $- |\hat{P}_{ss'}(a) - P_{ss'}(a)|$). A smaller gap indicates a more precise upper bound. As shown in Table 2, our algebraic bound achieves a significantly narrower gap across all goal depths. For instance, at $n = 100$ with $\delta = 0.01$, our tightness gap is approximately $0.35\%$, whereas the baseline statistical bound exhibits a gap of $5.31\%$. This represents a nearly 15-times improvement in precision, confirming that our method avoids the looseness typical of universal guarantees. Additionally, while the pass rate naturally decreases with $n$, the filter consistently retains a high-precision subset

*Table 2.* Recovered Error Tightness Gap Comparison (Lower is Better), $n \leq 400$

| $\Delta_f$ | 25 | | 50 | | 75 | | 100 | | 125 | | 150 | | 200 | | 300 | | 400 | |
|---|---|---|---|---|---|---|---|---|---|---|---|---|---|---|---|---|---|---|
| | **O** | **R** | **O** | **R** | **O** | **R** | **O** | **R** | **O** | **R** | **O** | **R** | **O** | **R** | **O** | **R** | **O** | **R** |
| 0.01 | **0.0124** | 0.0977 | **0.0059** | 0.0706 | **0.0042** | 0.0573 | **0.0035** | 0.0531 | **0.0021** | 0.0454 | **0.0023** | 0.0429 | **0.0020** | 0.0369 | **0.0020** | 0.0301 | **0.0016** | 0.0268 |
| 0.03 | **0.0144** | 0.0996 | **0.0080** | 0.0730 | **0.0064** | 0.0591 | **0.0053** | 0.0532 | **0.0047** | 0.0463 | **0.0048** | 0.0427 | **0.0041** | 0.0371 | **0.0036** | 0.0297 | **0.0032** | 0.0254 |
| 0.05 | **0.0168** | 0.1006 | **0.0113** | 0.0727 | **0.0091** | 0.0586 | **0.0075** | 0.0517 | **0.0074** | 0.0461 | **0.0070** | 0.0419 | **0.0067** | 0.0363 | **0.0058** | 0.0284 | **0.0056** | 0.0244 |
| 0.08 | **0.0217** | 0.1017 | **0.0156** | 0.0716 | **0.0134** | 0.0577 | **0.0121** | 0.0497 | **0.0126** | 0.0453 | **0.0116** | 0.0401 | **0.0111** | 0.0342 | **0.0107** | 0.0270 | **0.0101** | 0.0223 |
| 0.10 | **0.0268** | 0.1030 | **0.0197** | 0.0716 | **0.0177** | 0.0576 | **0.0159** | 0.0485 | **0.0162** | 0.0442 | **0.0154** | 0.0392 | **0.0144** | 0.0327 | **0.0140** | 0.0254 | **0.0137** | 0.0208 |
| 0.12 | **0.0322** | 0.1038 | **0.0247** | 0.0716 | **0.0224** | 0.0574 | **0.0200** | 0.0476 | **0.0206** | 0.0432 | **0.0195** | 0.0382 | **0.0190** | 0.0320 | **0.0188** | 0.0248 | **0.0184** | 0.0201 |
| 0.15 | **0.0422** | 0.1044 | **0.0324** | 0.0704 | **0.0300** | 0.0561 | **0.0275** | 0.0461 | **0.0275** | 0.0408 | **0.0268** | 0.0362 | **0.0261** | 0.0298 | 0.0255 | **0.0220** | 0.0249 | **0.0170** |

1  **Legend:** O = Ours; R = Richens et al. (2025).
2  Tightness gap is defined as $\text{Gap} = \text{Bound} - \mathbb{E}[|\hat{P}_{ss'}(a) - P_{ss'}(a)|]$. Lower values indicate tighter upper bounds.
3  All values are shown with 4 decimal places.
4  **Red highlights** denote the smaller value within each $(\Delta_f, n)$ pair; the other value is shown in blue.

of transitions that are strictly verified to be accurate.

To understand the source of the improvement by smaller failure rates $\delta$ shown in Figure 3, Figure 5 visualizes the error distributions. The histograms reveal that the uncertified transitions suffers from a heavy tail of high error transition estimates, where a significant portion of transitions exhibit large estimation errors. Our filtering algorithms systematically identify and excise these high error outliers. Consequently, the distribution of the certified transitions is tightly concentrated near zero. This confirms that the reduction in mean error observed previously is not an artifact of random fluctuation, but the result of structurally removing statistically unreliable estimates.

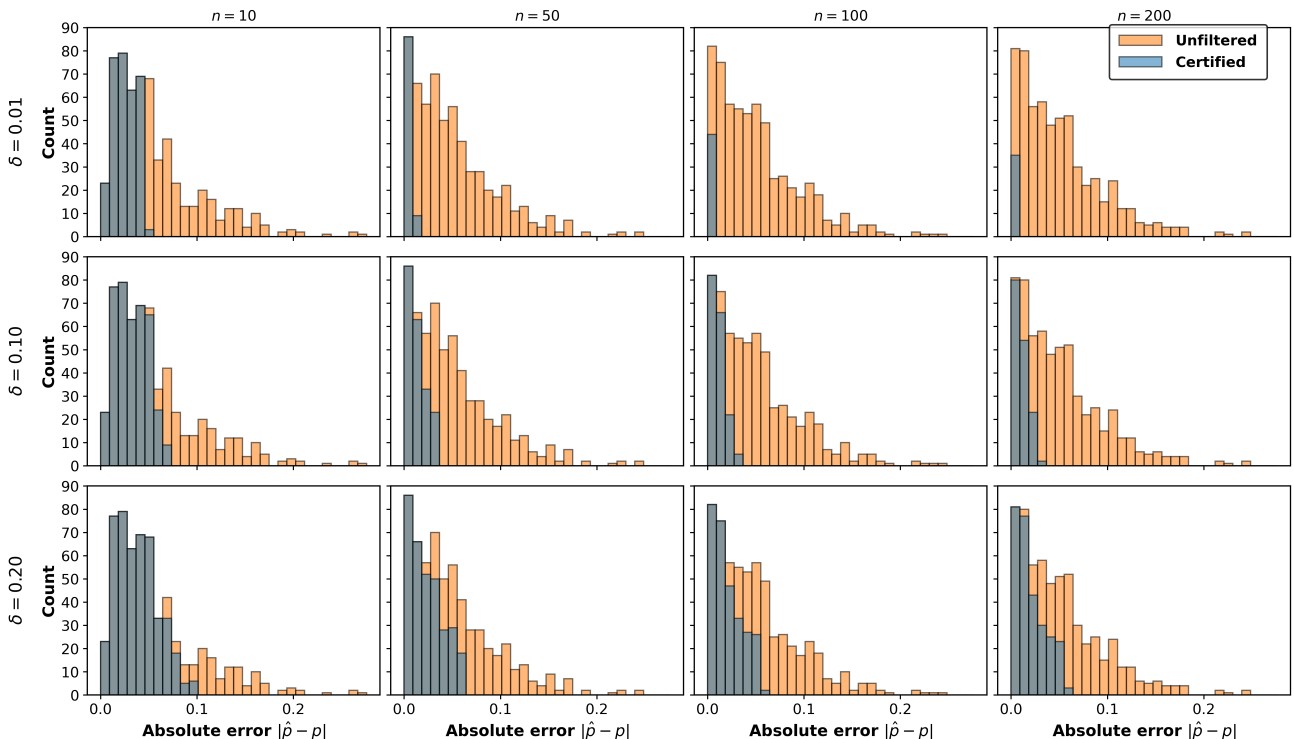

*Figure 5.* **Distribution of absolute estimation errors**. We compare the histograms of $|\hat{P}_{ss'}(a) - P_{ss'}(a)|$ for uncertified (orange) versus certified (blue) transitions across varying goal depths $n$ (columns) and failure rates $\delta$ (rows).

Furthermore, we evaluate our approach in a $100 \times 100$ maze environment and an Atari-based setting with goal depths ranging from $n = 10$ to $n = 100$. These results continue to support our main qualitative claims.

*Table 3.* Pass Rate of Filtering Algorithms, $n \leq 400$

| $\Delta_f$ | 25 | 50 | 75 | 100 | 125 | 150 | 200 | 300 | 400 |
|---|---|---|---|---|---|---|---|---|---|
| 0.01 | **26.0%** | **15.8%** | 12.5% | 7.3%[†] | 9.7%[†] | 8.2%[†] | 5.8%[†] | 3.2%[†] | 3.2%[†] |
| 0.03 | **30.3%** | **20.7%** | 17.0% | 12.3% | 13.3% | 11.3% | 10.0% | 9.0%[†] | 8.7%[†] |
| 0.05 | **35.7%** | **23.2%** | 21.0% | 18.0% | 17.8% | 16.5% | 14.3% | 14.2% | 13.0% |
| 0.08 | **43.5%** | **30.3%** | 28.7% | 25.2% | 22.8% | 23.2% | 21.5% | 20.0% | 20.0% |
| 0.10 | **46.0%** | **34.2%** | 31.3% | 29.2% | 27.0% | 27.0% | 26.5% | 25.0% | 24.7% |
| 0.12 | **49.7%** | **37.0%** | 34.3% | 33.5% | 30.5% | 31.0% | 29.3% | 27.5% | 27.2% |
| 0.15 | **53.5%** | **43.8%** | 40.7% | 40.2% | 37.8% | 37.7% | 36.0% | 35.0% | 35.0% |

1   **Legend:** Entries are pass rates (in %); higher is better.
2   **Blue cells** highlight short horizons ($n \in \{25, 50\}$) to emphasize short-horizon strength.
3   **Red cells** marked with [†] indicate pass rate $< 10\%$.
4   All values are shown with 1 decimal place.

*Table 4.* More detailed results of Atari-based environment ($|S| = 100$, $|A| = 5$, $T = 20000$). We report Pass Rate, Mean Error, Tightness (Ours), and Tightness (Richens) across filter thresholds $\Delta_f$ and horizons $n \in \{10, 20, 50, 100\}$.

| Pass Rate | 10 | 20 | 50 | 100 |
|---|---|---|---|---|
| 0.01 | **31.2%** | **22.5%** | 14.8% | 9.6% |
| 0.03 | **38.6%** | **28.1%** | 19.7% | 13.4% |
| 0.05 | **44.9%** | **33.8%** | 24.2% | 17.6% |
| 0.08 | **52.7%** | **40.5%** | 30.1% | 22.8% |
| 0.10 | **57.3%** | **45.2%** | 34.5% | 26.1% |

| Mean Error | 10 | 20 | 50 | 100 |
|---|---|---|---|---|
| 0.01 | **0.028** | **0.032** | 0.041 | 0.053 |
| 0.03 | **0.026** | **0.030** | 0.039 | 0.050 |
| 0.05 | **0.025** | **0.029** | 0.037 | 0.048 |
| 0.08 | **0.024** | **0.028** | 0.036 | 0.046 |
| 0.10 | **0.024** | **0.027** | 0.035 | 0.045 |

| Tightness (Ours) | 10 | 20 | 50 | 100 |
|---|---|---|---|---|
| 0.01 | **0.061** | **0.069** | 0.083 | 0.098 |
| 0.03 | **0.060** | **0.067** | 0.081 | 0.095 |
| 0.05 | **0.058** | **0.066** | 0.079 | 0.093 |
| 0.08 | **0.057** | **0.064** | 0.077 | 0.091 |
| 0.10 | **0.056** | **0.063** | 0.076 | 0.089 |

| Tightness (Richens) | 10 | 20 | 50 | 100 |
|---|---|---|---|---|
| 0.01 | **0.101** | **0.113** | 0.132 | 0.151 |
| 0.03 | **0.099** | **0.111** | 0.129 | 0.148 |
| 0.05 | **0.097** | **0.109** | 0.127 | 0.145 |
| 0.08 | **0.095** | **0.107** | 0.124 | 0.143 |
| 0.10 | **0.094** | **0.106** | 0.123 | 0.141 |

1   **Legend:** Each panel reports one metric as a function of filter threshold $\Delta_f$ and horizon $n$.
2   **Blue columns** highlight shorter horizons ($n = 10, 20$).
3   Higher is better for Pass Rate; lower is better for Mean Error and both Tightness metrics.
4   Our bound is consistently tighter than the Richens baseline across all settings.

Finally, we verify the convergence rate of the certified error in Figure 6 to verify our upper bounds. By fitting the empirical data to decay models, we compare our approach upper bounds provided by Richens et al. (2025). The results show that the certified error closely follows an $\mathcal{O}(\frac{1}{n}) + \mathcal{O}(\delta)$ scaling law (red curves), matching the theoretical fast rates derived in our main theorems. This is significantly faster than the $\mathcal{O}(\frac{1}{\sqrt{n}})$ rate (blue curves) typically associated with standard uniform convergence, providing strong empirical evidence that our method leverages the local structure of the dynamics to achieve superior sample efficiency.

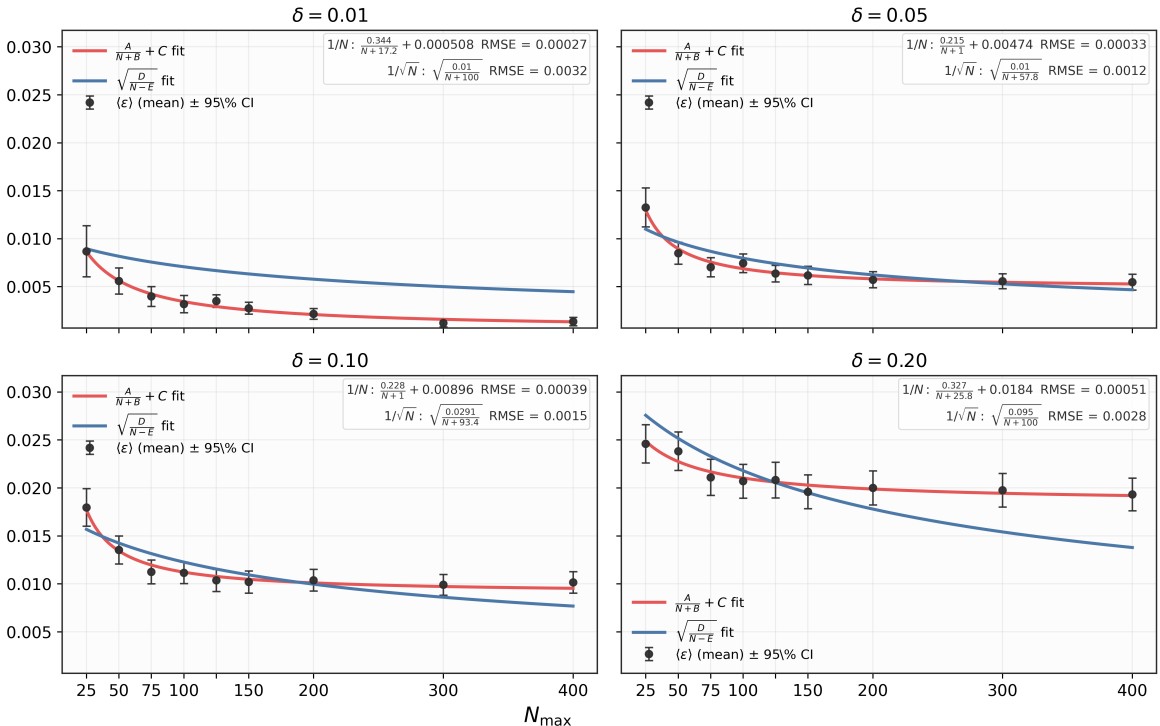

*Figure 6.* **Empirical scaling of certified estimation error**. We plot the mean estimation error $\langle \epsilon \rangle$ (black dots) with 95% confidence intervals against the goal depth $N_{\max}$ for varying $\delta$. We fit the data to two decay models: an $\mathcal{O}(1/n) + \mathcal{O}(\delta)$ (red curves) and $\mathcal{O}(1/\sqrt{n})$ (blue curves). The fitted equations and Root Mean Square Error (RMSE) are reported in each panel. The $\mathcal{O}(1/n) + \mathcal{O}(\delta)$ model yields a consistently lower RMSE, empirically verifying the fast convergence rates predicted by our theory.

**Case Study** We now provide the details for our case study. We consider a stochastic maze with $|\boldsymbol{S}| = 625$ states arranged on a $25 \times 25$ grid and $|\boldsymbol{A}| = 5$. The action set is $\boldsymbol{A} = \{\text{UP}, \text{DOWN}, \text{LEFT}, \text{RIGHT}, \text{STAY}\}$, where each action attempts to move to the corresponding adjacent cell (or remain in place for STAY). If the intended transition is blocked by a wall or exits the grid, the agent remains at the current state. Moreover, for any state $s$ and action $a \in \{\text{UP}, \text{DOWN}, \text{LEFT}, \text{RIGHT}\}$, the transition is local. That is, $P_{ss'}(a) = 0$ unless $s'$ is the intended adjacent cell of $s$ under $a$ or $s' = s$. We collect $N_{\text{samples}} = 30000$ transitions using a random walk exploration policy to build an empirical model $\widetilde{P}_{ss'}(a)$ for the agent. The agent then performs optimal model-based planning under $\widetilde{P}_{ss'}(a)$ for two compositional tasks (one is to pick up a key to unlock a door, and another is to pick up a pen to write a paper).

The concrete maze instances and the placement of task objects are shown in Figure 7a and Figure 7b. Black cells denote walls (or blocked cells). Using the ground-truth transition probabilities $P_{ss'}(a)$, we compute the transition estimate error $|\hat{P}_{ss'}(a) - P_{ss'}(a)|$ where $\hat{P}_{ss'}(a)$ is obtained through Algorithm 2 and Algorithm 3. To visualize both the recovery error and the effect of filtering algorithms, we use a complete recovery variant that outputs an estimate $\hat{P}_{ss'}(a)$ for every transition, regardless of whether it is certified. We use Algorithm 2 to recover all non-trivial transitions ($P_{ss'}(a) \in (0, 1)$) and we use the extended version of Algorithm 3 provided in Richens et al. (2025) to recover all trivial transitions ($P_{ss'}(a) \in \{0, 1\}$). The error $\epsilon(s)$ for each state $s$ is the total error of 5 actions:

$$\epsilon(s) := \sum_{a \in \boldsymbol{A}} \sum_{s' \in \mathcal{N}(s) \cup \{s\}} \left| \hat{P}_{ss'}(a) - P_{ss'}(a) \right|$$

where $\mathcal{N}(s)$ denotes the four-neighborhood of $s$ on the grid. For each cell representing a state $s$, blue indicates low total error between predictive world models and true dynamics and red indicates high total error.

As visualized in Figure 7a, when the task-relevant goals (the key and door) lie along trajectories whose constituent transitions are largely certified by our algorithms, the agent reliably reaches them with stable, directed behavior. In contrast, as shown in Figure 7b, when the goals (the pen and paper) require traversing regions that are largely absent from the filtered transition set, the agent's behavior becomes brittle. It repeatedly attempts blocked moves (colliding with walls), and fails to make consistent progress toward the objective.

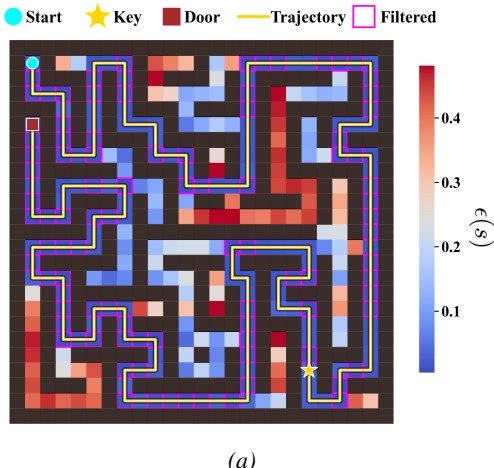

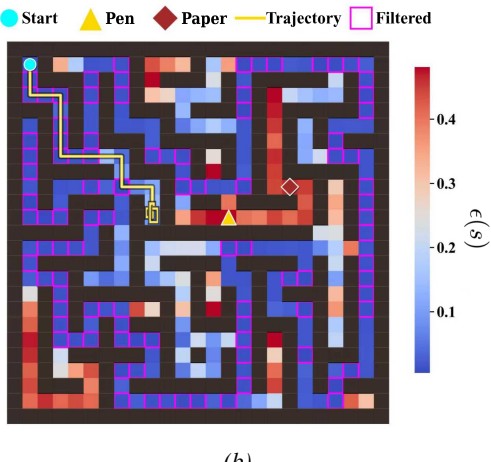

*(a)*                  *(b)*

*Figure 7.* **Certified regions depend on the task: key-door vs. pen-paper.** *(a)* Key-door task. *(b)* Pen-paper task. In both panels, cell color encodes a per-state aggregated dynamics recovery error $\epsilon(s)$, obtained by summing $|\hat{P}_{ss'}(a) - P_{ss'}(a)|$ over the five actions and the local next-state neighborhood. Magenta outlines mark certified transitions, and the yellow curve shows a representative trajectory from the start (circle) to the task objects (key/star and door/square in *a*; pen/triangle and paper/diamond in *b*).

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
