# OpenReview forum: "World Models in Pieces: Structural Certification for General Agents"
_ICML.cc/2026/Conference — ICML 2026 regular_

### Official Review · Reviewer_mCaj · 2026-03-04

**Soundness:** 3
**Presentation:** 3
**Significance:** 2
**Originality:** 3
**Overall Recommendation:** 5
**Confidence:** 4

**Summary:**

The authors examined general agents in big-world environments, where agents cannot be good at everything. They first show that the existing definition of general agents with uniform guarantees over all the goals is ןnvalid in complex environments and instead propose an alternative definition, which is a local view of general agents, whose capabilities are concentrated on specific parts of the environment. For this definition, they also proposed a transition-local certification framework that maps an agent's bounded goal conditioned performance on carefully constructed LTL goal sets to entry-wise accuracy guarantees on its internal world model, The core result is Theorem 3.2, which provides filtering algorithms (Algorithms 1+2), and proves a tight bound between the true transition probabilities and the agent's induced ones. In addition, the authors also introduced a fundamental limit in Theorem 3.3, proving that any attempt to infer a world
model for the agent outside of these bounds will result in an inaccurate model. Finally, experiments validate the framework, showing certified transitions form a conservative envelope of reliable regions.

**Compliance With Llm Reviewing Policy:**

Affirmed.

**Final Justification:**

I'm satisfied with the authors' response and raise my score

**Key Questions For Authors:**

please see weaknesses above

**Limitations:**

yes

**Strengths And Weaknesses:**

Points of strength:

S1: In Proposition 3.1, the authors show and prove that the “General Agents for Universal Goals” definition does not work in complex environments, because there are no agents that can guarantee good performance on all existing goals.

S2: The paper gives a better alternative definition and prove that it actually works in complex environments (unlike the previous definition). The authors suggest to replace universal evaluation with specific local checks and show that if an agent work well on those specific tasks, then they can prove that its internal model is good for those parts of the environment.

S3: Theorem 3.2 provides an upper bound on how close the agent’s estimated transition probabilities are to the true probabilities.

Weaknesses

W1: All theories and propositions of the papers rely on Assumption 2.1, which is that the theoretical framework relies on finite state and action spaces and full observability, but the authors did not discuss and provide how this framework might extend to continuous spaces, which are typical in the regimes on which this paper rely.

W2: The identification of bottleneck transitions is not discussed and defined formally in the paper, especially since the authors repeatedly refer to them. The authors could expand the discussion on how to identify or at least approximate bottleneck transitions, especially in large state spaces (big-world environments)

---

> ### Author Rebuttal · Authors · 2026-03-30
>
> **Response 4:** Thank you so much for your valuable feedback! We appreciate your positive assessment of the paper. We now address your questions below.
>
> **W4-1: Extension of Assumptions**
>
> **A4-1:** We acknowledge that the current theory is developed in a finite, fully observed setting. The purpose of this setting was to isolate the core structural question and keep the results as clean and transparent as possible. We agree that a natural next step is to extend the framework to more realistic settings, such as continuous spaces or partial observability.
>
> Further extensions are indeed possible, and we have thought carefully about how the framework could generalize beyond the current finite, fully observed setting. For continuous spaces, a natural next step is to replace a single discrete transition by a local region or partition of the state-action space, and ask whether bounded goal-conditioned performance can certify dynamics within a neighborhood or between adjacent cells, rather than a single entry $P_{ss'}(a)$. The target of the certification would then become a local transition map over a partition, likely under additional regularity assumptions such as smoothness or local Lipschitz continuity, so that entry-wise recovery is replaced by region-level or local-kernel certification.
>
> Under partial observability, the analogous extension is to replace state-level transitions by belief-state transitions or task-relevant latent transitions, and ask whether carefully designed probe goals can certify predictive competence over task-relevant latent dynamics rather than fully observed next states.
>
> Although we do not claim to solve these settings here, we view them as direct extensions of the same core idea: **replacing global whole-agent guarantees by certifiable local competence on the parts of the dynamics that matter for planning**. We will add this discussion explicitly in the revision.
>
> **W4-2: Definition of bottleneck transitions**
>
> **A4-2:** We thank the reviewer for this suggestion and agree that the terminology can be sharpened. In the existing literature, related ideas are often described informally as decision-critical steps, rather than introduced as a formal transition-level primitive [1][2]. We have discussed some prior works about bottleneck transitions in our "Related Work" section and we will make this distinction clearer below.
>
> In our paper, a bottleneck transition is **agent-specific**. For a fixed agent, we use our certification procedure to identify the transitions on which the agent demonstrates **reliable local planning capabilities**. We then deploy the agent only on tasks whose successful execution can be supported by this certified transition set. Relative to that agent and task family, we call a transition a bottleneck transition if successful executions frequently pass through it, or if failure on that transition destroys the downstream reachability required for task completion.
>
> Thus, bottleneck transitions are not assumed a priori and do not require a perfect bottleneck oracle. Rather, they arise from the interaction between task structure and the agent’s certified transition set. This is exactly the role of our framework: first identify which transitions are certifiably mastered, and then restrict deployment to tasks that rely on those certified pieces of the world model. We will clarify this terminology in both the main text and the related work discussion.
>
> Again, we are grateful for the reviewer's careful reading and rigorous evaluation of our work. Please let us know if there are any further questions or concerns.
>
> **References**
>
> [1] Zelei Cheng, Xian Wu, Jiahao Yu, Sabrina Yang, Gang Wang, and Xinyu Xing. RICE: Breaking Through the Training Bottlenecks of Reinforcement Learning with Explanation. arXiv preprint arXiv: 2405.03064, 2024.
>
> [2] Amirhossein Mesbah, Reshad Hosseini, Seyed Pooya Shariatpanahi and Majid Nili Ahmadabadi. Subgoal Discovery Using a Free Energy Paradigm and State Aggregations. arXiv preprint arXiv: 2412.16687, 2025.

---

> > ### Author Rebuttal · Reviewer_mCaj · 2026-04-02
> >
> > Thank you for your detailed response. I appreciate the clarification about bottleneck transitions emerging from the certification procedure, and your proposed extensions to continuous spaces and partial observability are convincing and important. Taken together, the approach you proposed and the edits address my concerns, so I believe that your promised revisions and future work will meaningfully strengthen the paper and especially its applicability to more realistic settings.

---

> > > ### Author Response · Authors · 2026-04-02
> > >
> > > We sincerely thank you for your constructive and supportive review. Please let us know if you have any further concerns.

---

### Official Review · Reviewer_YuYr · 2026-03-11

**Soundness:** 3
**Presentation:** 2
**Significance:** 2
**Originality:** 2
**Overall Recommendation:** 4
**Confidence:** 3

**Summary:**

A major question in AI is if agents can have Intelligence without representation, or if intelligent behavior always learns an (implicit) world model. The paper studies the question if general agents can be universal, proves that they can not, and offer improved algorithms that filter transitions. An experimental validation using a small maze is provided.

**Compliance With Llm Reviewing Policy:**

Affirmed.

**Final Justification:**

The author has properly clarified and answered my main concerns.

**Key Questions For Authors:**

- Perform large experiment, of large maze, and of other benchmarks. MuJoCo? Atari?
- Provide deeper explanation of why the filters work. Condense existing discussion to the essence.
- Provide small example to explain intuition.

**Limitations:**

Yes

**Strengths And Weaknesses:**

*Soundness*

The work appears sound. Two results are proved (the impossibility results of universal general agents, and structural certification). The proofs are present in the appendices, and they appear correct and make intuitive sense.

*Presentation*

The presentation is ok. The english is almost good, some sentences are not entirely correct (World model is crucial for a long…). Otherwise the text is fine and readable. There is appropriate theory, and a small experiment, many references. The explanation of why things work should be improved.

*Significance*

Significance of the work is where I am most hesitant. Clearly the major question about whether model free or model based is the way forward, or if model-free, do models get created, is an important question. This work provides a step this field, perhaps it is a small step, perhaps it is big enough to warrant acceptance. The appendices are certainly elaborate and the work appears sound enough.

It is a pity that only a small experiment is presented, and that no resutls on a bigger experiment are performed, and the analysis of the experiment is limited, no ablations.

*Originality*

The question on whether models are learned and how we should use this to achieve better algorithms is important, and the paper makes a step forward in this field.

---

> ### Author Rebuttal · Authors · 2026-03-30
>
> **Response 3:** We are very grateful for your constructive review and suggestions. We now respond to your concerns point by point below.
>
> **W3-1: Significance of our work**
>
> **A3-1:** We believe our work is significant for three main reasons.
>
> First, Proposition 3.1 is a **conceptual insight**: it formalizes why non-trivial uniform guarantees over universal goal sets are **uninformative** in reality, suggesting that evaluation of general agents should focus less on uniform certification of the entire agent [1][2], and more on identifying **planning-relevant transitions on which capabilities can be certified**. Second, Theorem 3.2 turns this into a **constructive alternative** by showing that bounded goal-conditioned performance can certify specific transitions, with concrete certification algorithms designed to evaluate bounded-performance transitions. Third, Theorem 3.3 shows that this localization is **necessary** rather than merely convenient: policy behavior alone cannot support comparably sharp world model recovery guarantees.
>
> Overall, this paper **reframes the design and evaluation of general agents**: away from universal notions of reliability, and toward certifiable, planning-relevant local capabilities that support more reliable compositional planning.
>
> **Q3-2: Experiments**
>
> **A3-2:** We thank the reviewer for this suggestion. We agree that broader empirical evidence is helpful, and we therefore include a $100\times 100$ maze environment and an Atari-based experiment with goal depth from $n=25$ to $n=100$ below, which continue to support the main qualitative claims of our theory. In our submitted paper, we compare our algorithms with [2] (see **Figure 2, 6, and Table 2**), which, to our knowledge, is the only prior work directly studying policy-query world model recovery. We also include an ablation experiment (see **Figure 3**) to show how the certification level affects the alignment of the internal world model.
>
> |$\delta$|Mean error|Pass rate|Tightness (Ours)|Tightness (Richens)|
> |---|---|---|---|---|
> |0.01|0.0090 $\to$ 0.0031|0.72 $\to$ 0.28|0.012 $\to$ 0.004|0.105 $\to$ 0.053
> |0.05|0.0111 $\to$ 0.0065|0.83 $\to$ 0.59|0.019 $\to$ 0.008|0.106 $\to$ 0.052
> |0.10|0.0132 $\to$ 0.0095|0.94 $\to$ 0.81|0.031 $\to$ 0.017|0.107 $\to$ 0.050
> |0.20|0.0147 $\to$ 0.0121|0.99 $\to$ 0.97|0.070 $\to$ 0.050|0.113 $\to$ 0.051
>
> Due to space limit, we only summarize the key quantitative trend here. We will incorporate the detailed setup and discussion in the revision. Significantly, as this paper is primarily theoretical, the role of the experiments is to verify the qualitative claims, rather than to pursue broad benchmark domination.
>
> **Q3-3: Explanation of why the filters work**
>
> **A3-3:** We provide a simple intuitive explanation: **the filter constructs a transition-isolating family of probe goals whose pass/fail pattern induces a switching threshold around the true value $P_{ss'}(a)$**. By discretizing [0,1] into $n+1$ levels, the algorithm localizes $P_{ss'}(a)$ between neighboring grid points. This gives an ideal resolution of order $1/(n + 1)$ in the optimal case, and hence an error scale of roughly $1/2(n+1)$. The role of failure rate $\delta$ is to blur this threshold through bounded sub-optimality, yielding the additional $\mathcal{O}(\delta)$ term in Theorem 3.2. We will include the intuition into our main text in revision.
>
> **Q3-4: Example of our idea intuition**
>
> **A3-4:** We use our **key-door case study** to illustrate the intuition behind our framework.
>
> In a big-world regime, a general agent may perform reliably on some parts of the environment but not on others. Thus a single global judgment over all tasks can be misleading: it may overstate capabilities on tasks that depend on weak parts of the agent’s behavior, or understate capabilities on tasks built from parts it handles consistently well. Our framework instead asks which parts of the environment support certifiable planning capabilities. In **Figure 7**, the key-door task is reliable since the agent's internal world model is aligned with the parts of the environment required for that task, whereas the pen-paper task is fragile because it depends on parts where this consistency is absent. This illustrates the core idea of the paper: evaluation should identify which parts of the environment support certifiable planning capabilities, rather than seek a single uniform verdict over all tasks.
>
> We hope our response addresses your concerns. We want to thank you once again for your constructive review. Please let us know if you have any additional concerns or suggestions.
>
> **References**
>
> [1] Jonathan Richens and Tom Everitt. Robust agents learn causal world models. In The Twelfth International Conference on Learning Representations, 2024.
>
> [2] Jonathan Richens, David Abel, Alexis Bellot, and Tom Everitt. General agents contain world models. arXiv preprint arXiv:2506.01622, 2025.

---

> > ### Author Rebuttal · Reviewer_YuYr · 2026-04-03
> >
> > Thank you for the clarifying comments. All my concerns have been addressed, and I will update my score accordingly.

---

> > > ### Author Response · Authors · 2026-04-03
> > >
> > > We sincerely thank you for your thoughtful and encouraging feedback. We would be happy to clarify any remaining concerns.

---

### Official Review · Reviewer_UKo4 · 2026-03-13

**Soundness:** 3
**Presentation:** 3
**Significance:** 2
**Originality:** 3
**Overall Recommendation:** 4
**Confidence:** 2

**Summary:**

The paper introduces a new idea to filter out specific transitions in order to narrow down the impact of errors in the world model of an agent on the performance. They show that an agent cannot universally guarantee its ability to plan in complex environments. They then introduce “filtering algorithms” which can

**Compliance With Llm Reviewing Policy:**

Affirmed.

**Key Questions For Authors:**

See above.

**Limitations:**

See above.

**Strengths And Weaknesses:**

# Strengths:

I believe the argument for the filtering and certification are well made. Although the work is not in an area of my expertise, I can see the experiments corroborate the theoretical argument very well.

# Weaknesses:

I believe Proposition 3.1 is slightly straightforward. This not to question the contributions, but reading the proof this is a consequence of $\pi$ being non-optimal and construction of a $\psi_{fail}$ such that $(1 - \gamma)^N < 1 - \delta$, where $\gamma$ is the margin by $\pi$ is sub-optimal. I don't see how this is a “a fundamental impossibility of seeking universal guarantees in complex environments”. I believe it has to be interpreted in light of the optimality gap of $\pi$ and $N$. While I see the efficacy of Theorem 3.2 and the subsequent algorithm resulting from it, it is hard for me to grasp the usefulness of the impossibility result.

The other issue is that the algorithms and their explanation are in the appendix and leave some unanswered questions. I would expect them to be explained in the main body.

**Other smaller issues:**

line 100 left side: what is a filtering algorithm? This has not been introduced in the text yet. You use it again in Theorem 3.2 but demote them to the appendix.

Assumption 2.1: what do you mean by reaching $s’$ from $s_0$? Non-zero probability or with probability 1 with a finite set of actions?

Line 95 right side: I believe the notation should be $\mathbf{g} \subseteq \mathbf{S} \times \mathbf{A}$ instead of a union. Your goal, if it were just a single tuple, is not either a state or an action but instead a tuple and therefore the set of goals is from the cross product of state and action sets.

---

> ### Author Rebuttal · Authors · 2026-03-30
>
> **Response 2:** We would like to express our sincere gratitude for your positive and careful feedback. We respond to your concerns point by point below.
>
> **W2-1: The conceptual usefulness of Proposition 3.1**
>
> **A2-1:** Thank you for this insightful comment. We agree with your key point. Proposition 3.1 should be interpreted as a **conceptual insight**, rather than the main technical result of the paper.
>
> Your intuition is exactly right: **the dependence of failure rate on the policy’s sub-optimality gap and the goal depth shows how even a fixed local gap can be amplified by composition, making uniform certification over all goals increasingly uninformative in the big-world regime.** Under the classical definition, a general agent is bounded and goal-conditioned [1]. Proposition 3.1 shows, however, that for any non-optimal agent, must fail this requirement on some goal set, thereby precluding universal certification. This is precisely what motivates our shift in Theorem 3.2 from universal guarantees to transition-local structural certification. In other words, Proposition 3.1 is intended to justify why the central question should not be whether one can certify **an entire general agent uniformly over all tasks** [1], but rather which **planning-relevant transitions can be certified** from bounded goal-conditioned performance.
>
> We can again consider the “key-door” case study in our paper. Proposition 3.1 suggests that the agent only needs to exhibit strong planning capability on certain transitions (see **Figure 7**) that are critical for reaching the goal—in our setting, picking up the key and opening the door, rather than uniformly across the full environment.
>
> We agree the role of Proposition 3.1 can be stated more explicitly and carefully in the main text, and we will revise our presentation of Proposition 3.1 accordingly.
>
> **W2-2: Algorithms and explanations**
>
> **A2-2:** We will include a simple version of the algorithm in the main text as follows, together with a more detailed discussion of the algorithms. For example, **Algorithms 1 and 2** are **the first policy-query algorithms designed to certify which transitions of a black-box general agent support reliable long-horizon planning**. Our algorithms could support trustworthy deployment of the black-box agent on tasks, whose success depends on those certified transitions, with its planning competence on those task-relevant parts supported the algorithms.
>
> **Simple certification procedure for $(s,a,s')$**
>
> **Input:** Candidate transition: $(s, a, s')$; certification parameters $(\delta, n)$; anchor $p = P_{ss'}(a)$; black-box goal-conditioned policy $\pi$.
>
> **Step 1:** Build a small family of probe goals indexed by $ r = \\{ 0, 1, \cdots, n-1 \\} $. Each probe repeatedly tests the same goal $s \xrightarrow{a} s'$ for $n$ times, while contrasting it with an alternative first action $b \neq a$.
>
> **Step 2:** For each probe goal, query the black-box agent and record whether its preferred first action is $a$ or $b$.
>
> **Step 3:** As the probe index $r$ increases from $0$ to $n - 1$, agent should exhibit a threshold behavior: for smaller $r$, action $b$ is preferred; after a switching point, action $a$ becomes preferred.
>
> **Step 4:** Check whether the observed switching point $r$ and transition probability $p$ is within the allowed $(\delta, n)$-band: $p \in \left[ \frac{(r + 1)(1 - \delta)}{n + 1 - (r + 1)\delta}, \frac{r + 1}{n + 1 - \delta(n - r)} \right]$. If yes, output $\mathrm{Cert}(s,a,s')=\mathrm{True}$. Otherwise, output $\mathrm{Cert}(s,a,s')=\mathrm{False}$.
>
> **Step 5:** If the transition is certified, convert the switching point into the induced estimate $\hat{P}_{ss'}(a)$.
>
> **Typo issues**
>
> **A2-3:** We sincerely appreciate your careful reading! The term "filtering algorithm" indicates the same thing as "certification algorithm", namely Algorithm 1 and Algorithm 2, which certify whether a transition is $\delta, n$ bounded. We will refer these algorithms unitedly as "certification algorithm" and formally define it before Theorem 3.2 in the revised version.
>
> Assumption 2.1 (iii) should be interpreted as "with probability 1 with a finite sequence of actions". We agree this should be clarified in the text and we would add it in the revision. This assumption is mainly a technical reachability condition used to simplify the proof constructions, and this will not impact on our final conclusion and main ideas of this paper.
>
> Line 95: You are definitely right, and we will modify it to **g** $\subset$ **S** $\times$ **A** in the revised version.
>
> Thank you once again for your support and your careful review. We hope this addresses your concerns, and we would be happy to clarify any further question.
>
> **References**
>
> [1] Jonathan Richens, David Abel, Alexis Bellot, and Tom Everitt. General agents contain world models. arXiv preprint arXiv:2506.01622, 2025.

---

> > ### Author Rebuttal · Reviewer_UKo4 · 2026-04-05
> >
> > Thank you for clarifying the relevance of Proposition 3.1 and describing the simplified algorithm. I will keep my current score as I am not an expert in this field. The authors have resolved all my comments and concerns.

---

> > > ### Author Response · Authors · 2026-04-05
> > >
> > > Thank you very much for your positive and careful review. Please let us know if you have any further suggestions or concerns.

---

### Official Review · Reviewer_C7Sw · 2026-03-13

**Soundness:** 3
**Presentation:** 2
**Significance:** 2
**Originality:** 2
**Overall Recommendation:** 3
**Confidence:** 3

**Summary:**

This paper takes a deeper look at the extent to which safety related guarantees are still possible in the big world scenario where world models must be incomplete by definition. Towards this end, the paper proposes a novel certification procedure that acts on a transition-local level. In proposition 3.1 the authors provide an impossibility result demonstrating that agents cannot jointly achieve all goals to a fixed failure rate. This then leads them to propose a transition level probe that certifies world model performance on bottleneck transitions and demonstrates that this process leads to a tight error bound. The authors provide filtering algorithms to isolate specific important transitions.

**Compliance With Llm Reviewing Policy:**

Affirmed.

**Key Questions For Authors:**

Q1: Can you make a simple bullet point list of the main contributions?

Q2: What conceptual contribution does Proposition 3.1 make over the no free lunch theorem?

Q3: Can you provide a simple algorithm describing the transition certification process?

Q4: Can you explain the greater significance of the result you present in Theorem 3.3? I have tried to understand it a few times, but feel like I am failing to understand the point the authors are trying to highlight.

**Limitations:**

Yes.

**Strengths And Weaknesses:**

I really like the motivation of this paper and feel that the authors may really be on to something. However, I feel that it is very difficult for a general audience to absorb as currently written, which unfortunately will limit its impact. The problem is that the story is just too complex and too long to follow in my opinion. For example, even when I was trying to write the summary of this paper, I found just trying to go through all the claimed contributions exhausting. I feel like the paper needs more focus and the important parts need to be more fleshed out in the main text. This is clearly evident during the explanation of the contributions which is presented as half a page of free form writing rather than a well structured bullet point or numbered list.

Specifically when it comes to the presented theory, I feel like the authors do not use their space effectively. As the authors point out in the conclusion, Proposition 3.1 is closely related to the no free lunch theorem. While there are some technical contributions here, I don't think this result really makes a conceptual contribution to readers interested in the motivation of this paper. The authors then go on to the certification process, but it is kept very mathematical. It is very confusing when reading the text concretely how a transition would be certified. Maybe this can be clarified through a simple algorithm? Additionally, if Algorithms 1 and 2 are considered a contribution, the level of discussion about them in the main text is really unacceptable. Not only are they in the appendix, but they are not described on any level in the main text. Meanwhile, they seem to be key aspects of the main theoretical result in Theorem 3.2 and foundational to the experiments. Finally, I find it very difficult to understand why the authors are emphasizing Theorem 3.3.

---

> ### Author Rebuttal · Authors · 2026-03-30
>
> **Response 1:** Thank you so much for your constructive comments, which help us improve the presentation of the paper. We address your questions one by one below.
>
> **Q-1.1: Can you make a simple bullet point list of the main contributions?**
>
> **A-1.1:** Main contributions:
>
> 1. We show universal certification is unattainable in the big-world regime, and we are the first to show planning capability must instead be certified on task-relevant transitions.
>
> 2. We are the first to give a constructive transition certification procedure, together with a policy-query method and tight theoretical guarantees.
>
> 3. We show limitations of policy-query algorithms for world model recovery.
>
> | Result | Contribution
> |---|---|
> | **Proposition 3.1** | shows why universal certification is the wrong target in the big-world regime. (see **A-1.2**)|
> | **Theorem 3.2 + Algorithm 1 / 2** | give the first constructive transition-local certification procedure for black-box agents with guarantees. (see **A-1.3**)|
> | **Theorem 3.3** | shows this localization is necessary for identifiable recovery, and that uncertified transitions cannot generally be recovered at a better rate. (see **A-1.4**)|
>
> **Q-1.2: What conceptual contribution does Proposition 3.1 make over the no free lunch theorem?**
>
> **A-1.2:** We appreciate this constructive feedback. We view Proposition 3.1 as a no-free-lunch result for general agents. Under the classical definition, a general agent is bounded and goal-conditioned [2], yet Proposition 3.1 shows that for any non-optimal agent, one can always construct a set of goals on which this requirement fails, ruling out universal certification. This identifies the core failure mode in realistic scenarios: composition amplifies a fixed sub-optimality gap, so uniform certification over all tasks is not just hard, but the wrong target. In this sense, our result complements prior work showing that sufficiently strong global robustness can imply learned world models [1]. It also motivates Theorem 3.2: **the right question for general agents is not whether the whole agent can be certified uniformly, but which planning-relevant transitions can be certified from bounded goal-conditioned performance.**
>
> Our "key-door" case study illustrates this: the agent only needs reliable planning capability on specific transitions (see **Figure 7**), rather than uniformly across the full environment. We present Proposition 3.1 primarily as a conceptual insight that motivates the transition-local perspective developed in Theorem 3.2. We would further improve this discussion and state its role more explicitly.
>
> **Q-1.3: Contributions of Algorithms 1 and 2. Can you provide a simple algorithm describing the transition certification process?**
>
> **A-1.3:** Contributions: Algorithms 1 and 2 are, to our knowledge, **the first policy-query algorithms designed to evaluate and certify which transitions of a black-box general agent support reliable long-horizon planning capabilities**. By finding and certifying such transitions, our algorithms enable trustworthy deployment of the black-box agent on tasks whose success depends on those certified transitions, with formal guarantees on its planning capability over those task-relevant parts. We will state this role more explicitly in the revision.
>
> Additionally, we will move a simple algorithm pseudocode to the main text. Briefly, the procedure is: 1. construct transition-isolating probe goals, 2. query the black-box agent on them, and 3. use the pass/fail switching pattern to certify the target transition. Since Reviewer UKo4 raises a similar question, please refer to our response **A2-1** to Reviewer UKo4 for full details due to space limits. We formally discuss the implications of the algorithms after Theorem 3.2 and again in the Conclusion section. We will also provide a brief intuitive explanation of why the certification algorithm works. Please see our response **A3-3** to Reviewer YuYr for full details.
>
> **Q-1.4: Can you explain the greater significance of the result you present in Theorem 3.3?**
>
> **A-1.4:** Theorem 3.3 shows that transition-local certification is not only sufficient but necessary for **tight, identifiable recovery**. It contrasts with [2], which recovers world models from policies under universal bounded goal-conditioned assumptions. We show that without certified transition-isolating probes, black-box policy behavior is underdetermined, so no comparably sharp transition-level guarantee is possible. Hence local certification is needed in deployable black-box settings.
>
> Thank you once again for your valuable feedback. Please let us know if you have any further concerns.
>
> **References**
>
> [1] Jonathan Richens and Tom Everitt. Robust agents learn causal world models. In The Twelfth International Conference on Learning Representations, 2024.
>
> [2] Jonathan Richens, David Abel, Alexis Bellot, and Tom Everitt. General agents contain world models. arXiv preprint arXiv:2506.01622, 2025.

---

> > ### Author Rebuttal · Reviewer_C7Sw · 2026-04-01
> >
> > I don't feel like the spirit of my questions were really addressed in the rebuttal.
> >
> > The comment "we view Proposition 3.1 as a no-free-lunch result for general agents" doesn't make any sense to me on the surface. Isn't the no free lunch theorem directly addressing the feasibility of constructing a general agent?
> >
> > The algorithm provided in the response to the other reviewers is both missing details on how various steps work and also lacks any explanation of the intuition behind why certain things are done as part of the certification. Hopefully with more space this can be clarified.
> >
> > I still did not find the explanation of Theorem 3.3 accessible. The comparison with [2] also just leaves me even more confused. You can't just throw around terms like "under universal bounded goal-conditioned assumptions" and expect this to be digestible.

---

> > > ### Author Response · Authors · 2026-04-01
> > >
> > > We sincerely thank you for your thoughtful follow-up questions. We would like to clarify your concerns below.
> > >
> > > **Q1: Proposition 3.1 vs. No free lunch theorem**
> > >
> > > Thank you for your follow-up question. To clarify the conceptual contributions, we summarize the differences between Proposition 3.1 and NFL theorem below.
> > >
> > > At a high level, NFL is about the impossibility of universal **algorithmic superiority** over **all tasks**. Proposition 3.1 is different: we **do not use it to argue that constructing a general agent is infeasible**, but it shows even for a fixed policy in a fixed structured environment, **one cannot require a single non-trivial uniform guarantee** (see Eq.(1) in main text) to hold over all goals.
> > >
> > > |Aspect|No-Free-Lunch Theorem|Proposition 3.1|
> > > |---|---|---|
> > > |Objectives differ|A **universally superior algorithm** does not exist in average performance over all tasks.|**A single non-trivial uniform guarantee** does not exist over the whole goal space.|
> > > |What causes failure?|Global symmetry across $F$|**A specific compositional goal** can invalidate the whole global guarantee.|
> > > |Conceptual contributions|No **algorithm** can be uniformly superior when averaged over all tasks.|No **single global guarantees** over all compositional goals can serve as a meaningful global certification metric for general agents.|
> > >
> > > **Q2: Algorithm details and intuition**
> > >
> > > Below we summarize the certification procedure step by step and explain the intuition behind each step.
> > >
> > > At a high level, we repeatedly ask the agent to choose between goals that require fewer vs. more successful occurrences of the same target transition. The point where its preference flips tells us the model-implied probability range for the target transition.
> > >
> > > |Algorithm step|Why it works|
> > > |---|---|
> > > |1.**Goal construction.** Construct a family of probe goals $\psi$ by repeating $\\{s \xrightarrow{a}s'\\}$ $n$ times.|This makes the optimal success probability dependent only on transition probability $P_{ss'}(a)$, partitioning the range of the transition probability over $[0,1]$ into $(n+1)$ intervals.|
> > > |2. **Policy query.** For each index $r$, we construct two different actions and corresponding goals: choosing $a$ corresponds to a goal $\psi_a\\{r,n\\}$ with exactly $r$ successful occurrences of $s \xrightarrow{a} s'$, while choosing $b \neq a$ corresponds to a competing goal $\psi_b\\{r+1,n\\}$ with exactly $r+1$ such success occurrences in $n$ repetitions. Sweep $r= 0,1,\cdots,n-1$ and query the agent to choose a first action $a_r\in \\{a,b\\}$ and the corresponding goal.|An optimal agent always selects a goal with higher success probability. Thus, the agent’s choice of $a_r$ reveals whether its internal world model behaves as if the target transition probability were closer to the $r$-success side or the $(r+1)$-success side.|
> > > |3. **Certification.** For each $r$, if $a_r = a$, update $p_{max} \leftarrow \min\\{p_{max},\frac{r+1}{n+1-\delta(n-r)}\\}$; else, update $p_{min} \leftarrow \max\\{p_{min},\frac{(r+1)(1-\delta)}{n+1-(r+1)\delta}\\}$. We check whether $P_{ss'}(a) \in [p_{min}, p_{max}]$. If yes, output $\mathrm{Cert}(s,a,s')=\mathrm{True}$. Otherwise, the transition is not certified.|Each choice of $a_r$ reveals a preference between the two competing goals. The agent's preference is equivalent to whether the internal world model is closer to the $\frac{r}{n+1}$ side or the $\frac{r+1}{n+1}$ side, shrinking the feasible range of $P_{ss'}(a)$. Thus, the full preference pattern across all $r$ progressively narrows $[p_{min},p_{max}]$, and certification succeeds only if the true transition probability remains inside the final interval.|
> > > |4.**Estimate.** Record the first switching point $r*$ where the preferred first action $a_r$ changes from $b$ to $a$. Estimate internal world model $\\hat{P}_{ss'}(a)=\frac{r^*+0.5}{n+1}$.|The switching point $r*$ localizes $\\hat{P}_{ss'}(a)$ on one grid cell, and we can estimate it by the midpoint $\frac{r^*+0.5}{n+1}$. The sub-optimality level $\delta$ will result in $\mathcal{O}(\delta)$ term. This is why the algorithm can recover an interval estimate for internal world model with bounded error.|
> > >
> > > **Q3: Explanation of Theorem 3.3**
> > >
> > > Thank you for pointing this out. Theorem 3.3 explains why certification is necessary in the first place, which rules out policy-only recovery as a reliable route for recovering world models.
> > >
> > > The significance of Theorem 3.3 is simple but non-trivial: policy observations alone are insufficient to identify the true world model. A method that tries to recover a model from observed behavior can return a model that explains the same choices while still being wrong about the true transition probabilities. This matters because our question is not whether the recovered model can rationalize the agent’s actions after the fact, but whether the predictive model is correct on specific transitions.
> > >
> > > Thank you again for your careful review. Please tell us if you have further concerns.

---

### Decision · Program_Chairs · 2026-04-30

**Decision:**

Accept (regular)

**Comment:**

This paper appears to have a novel take on basic and old question in AI: the role of models. This perspective was universally appreciated by the reviewers. It would be nice if there were more experiments, though the authors do mention some additional experience in their rebuttal.

The one negative review is not extremely negative. The main concern seems to be presentation issues. The authors made some effort to address that, though their final response was not acknowledged.